# Human RAD52 stimulates the RAD51-mediated homology search

Ali Akbar Muhammad[1,*], Clara Basto[1,*], Thibaut Peterlini[2,3] , Josée Guirouilh-Barbat[4], Melissa Thomas[2,3], Xavier Veaute[5], Didier Busso[5] , Bernard Lopez[4] , Gerard Mazon[1] , Eric Le Cam[1], Jean-Yves Masson[2,3], Pauline Dupaigne[1]

Homologous recombination (HR) is a DNA repair mechanism of double-strand breaks and blocked replication forks, involving a process of homology search leading to the formation of synaptic intermediates that are regulated to ensure genome integrity. RAD51 recombinase plays a central role in this mechanism, supported by its RAD52 and BRCA2 partners. If the mediator function of BRCA2 to load RAD51 on RPA-ssDNA is well established, the role of RAD52 in HR is still far from understood. We used transmission electron microscopy combined with biochemistry to characterize the sequential participation of RPA, RAD52, and BRCA2 in the assembly of the RAD51 filament and its activity. Although our results confirm that RAD52 lacks a mediator activity, RAD52 can tightly bind to RPA-coated ssDNA, inhibit the mediator activity of BRCA2, and form shorter RAD51-RAD52 mixed filaments that are more efficient in the formation of synaptic complexes and D-loops, resulting in more frequent multi-invasions as well. We confirm the in situ interaction between RAD51 and RAD52 after double-strand break induction in vivo. This study provides new molecular insights into the formation and regulation of presynaptic and synaptic intermediates by BRCA2 and RAD52 during human HR.

## Introduction

Homologous recombination (HR) is an evolutionarily conserved process that plays a pivotal role in genome stability, diversity, and plasticity. HR is indeed a key repair pathway able to faithfully repair DNA damages including double-strand breaks (DSBs) and DNA gaps by copying the error-free information from the template DNA normally present in the sister chromatid (1, 2, 3). Defects in HR are associated with genetic instability, chromosomal aberrations, carcinogenesis, and cell death (4). The HR pathway is initiated by

the formation of single-stranded DNA (ssDNA) through the resection of double-stranded DNA (dsDNA) from a DSB end or the enlargement of an ssDNA gap. The ssDNA is initially covered by the replication protein A (RPA), and with the help of a number of protein mediators, the recombinase RAD51 can displace RPA from this ssDNA to form a presynaptic filament able to search and pair with the homolog dsDNA donor giving rise to the formation of joint molecules known as synaptic intermediates. The D-loop is the stable joint molecule formed upon invasion of the homologous dsDNA donor by the presynaptic RAD51 nucleofilament after the alignment of the complementary strands and subsequent displacement of the third strand. The invading strand then serves as a primer to start synthesis within the D-loop enabling the recovery of the information lost at the original break point. In the postsynaptic steps, the different synaptic intermediates would be resolved through alternative subpathways involving multiple helicases and structure-selective nucleases (5).

The assembly and regulation of the RAD51 filament on DNA are crucial for the proper formation of synaptic intermediates and their outcome. It is also now well established that RAD51 and some partners play additional roles in the protection of DNA from nuclease attack and extensive resection at DSBs and during replication (6, 7). RAD51 is an ATP-modulated protein that forms right-handed helical filaments on DNA (mostly ssDNA) (8) in which the DNA is stretched non-uniformly by 150% with a gap for every three nucleotides, each triplet following the B-shape of DNA (9). Human RAD51 also polymerizes and binds stably to dsDNA as efficiently as to ssDNA (10). The ssDNA-bound filaments form faster than those polymerized at dsDNA, but dsDNA filaments are stable once formed (11, 12, 13). Although the importance of dsDNA-bound nucleofilaments remains unclear, their accumulation in the absence of regulators like RAD54 indicates that they may be toxic intermediates if not timely disassembled (14, 15, 16, 17).

The HR homology search and strand exchange processes rely on the remarkable structure and properties of this filament. RAD51 has two DNA-binding sites: site I oriented inside the filament binds to

[1]Genome Integrity and Cancers UMR 9019 CNRS, Université Paris- Saclay, Gustave Roussy, Villejuif Cedex, France    [2]Genome Stability Laboratory, CHU de Quebec Research Center, HDQ Pavilion, Oncology Axis, Quebec City, Canada    [3]Department of Molecular Biology, Medical Biochemistry and Pathology, Laval University, Quebec City, Canada    [4]INSERM U1016, UMR 8104 CNRS, Institut Cochin, Equipe Labellisée Ligue Contre le Cancer, Université de Paris, Paris, France    [5]CIGEx Platform, INSERM, IRCM/IBFJ CEA, UMR Stabilité Génétique Cellules Souches et Radiations, Université de Paris and Université Paris-Saclay, Fontenay-aux-Roses, France

Correspondence: pauline.dupaigne@gustaveroussy.fr
*Ali Akbar Muhammad and Clara Basto contributed equally to this work

 

ssDNA, and site II allows to transiently contact the dsDNA donor. The filament likely facilitates base-flipping of triplet units, thereby facilitating homology probing and recognition by triplet base increments (9, 18, 19, 20, 21, 22). The homology probing has been shown to be based on tracts of eight-nucleotide microhomology and transient interactions between stretched single-stranded DNA within the filament and bases in a locally melted or stretched DNA duplex (21, 22, 23, 24, 25, 26, 27). The interaction between the nucleoprotein filament on ssDNA and the duplex DNA donor results in their incorporation into a three-stranded intermediate, the synaptic complex (SC), also known as a paranemic joint (28, 29, 30). Two types of SCs have been described: those in which DNA strand pairing is maintained by RAD51 (sensitive to deproteinization), and those in which the invading ssDNA of the filament and the complementary strand of the dsDNA donor are aligned and intertwined to form a new heteroduplex (resistant to deproteinization) (31, 32). In the latter case, the heteroduplex and the displaced strand form the displacement loop (D-loop), an important HR intermediate required to prime DNA synthesis by the 3′ OH of the invading strand in the heteroduplex (33). Many studies have contributed to a better understanding of homology search and the D-loop dynamics; however, the mechanistic steps and the specific roles of associated RAD51 partners leading to the RAD51-mediated SC are incompletely characterized.

In humans, many RAD51 partners have been identified as playing roles in the filament formation, its architecture, and its activity in searching for homology and the handling of the subsequent D-loop. RAD51 mediators are proteins that help filament assembly and stabilization, either by accelerating RAD51 nucleation on RPA-ssDNA or by decelerating its dissociation from ssDNA. BRCA2 and the BCDX2 complex formed by RAD51 paralogs mediate the nucleation of RAD51 filaments onto ssDNA covered by RPA (34, 35, 36, 37), whereas some other RAD51 paralogs have been shown to bind and remodel the presynaptic filament to stabilized and flexible conformation (38). BRCA2 also directly binds RAD51 through BRC repeats (under substoichiometric conditions) and selectively targets RAD51 to ssDNA, thus reducing non-productive interactions with dsDNA (34, 39, 40, 41). In contrast, the binding of the BRCA2 TR2 C-terminal domain to RAD51 stabilizes RAD51 binding to dsDNA, even in the presence of BRC4, thus promoting DNA protection against nuclease activities (42). A BRC peptide was also shown to intercalate between RAD51 protomers within the filament, inhibiting RAD51 ATPase activity and thereby suppressing RAD51 release from DNA (43). Finally, a postsynaptic function of BRCA2 has been proposed involving the inhibition of RAD51 excess–mediated D-loop dissociation, highlighting a role of homeostasis between RAD51 and BRCA2 as an important factor for HR in mammalian cells (44, 45). In the yeast *Saccharomyces cerevisiae*, Rad52 is identified as the main HR mediator through Rad51 filament nucleation catalysis (46, 47, 48) and has been shown to directly intercalate into the presynaptic filaments by forming with Rad51 mixed filaments rendering them more resistant to the helicase Srs2 antirecombinase activity (49). Human RAD52 shares a number of biochemical properties with its yeast counterpart, including the formation of ring oligomers (50, 51, 52), the ability to catalyze the annealing of complementary strands, and its potential cooperation with RAD51 in strand exchange activity (53, 54, 55). hRAD52

has also been shown to have functions independent of RAD51 in alternative DSB repair pathways, including single-strand annealing (SSA) and break-induced replication, and to promote DNA synthesis during replication stress (56, 57). Although unlike its yeast homolog, hRAD52 has not been shown to mediate RAD51 filament formation, its cooperation with RAD51 has been observed in certain DNA damage repair contexts (58, 59). Moreover, in light of the lack of strong phenotypes for the RAD52 mutants in vertebrates (60, 61), RAD52 was considered to play a dispensable role, somehow redundant with other players, but becoming essential in the absence of key recombination proteins, including BRCA2, and most RAD51 paralogs, as highlighted by the synthetic lethality conferred by its mutation combined with that of these proteins (58, 62). It is still under debate whether RAD52 plays any role in the early stages of HR, including RAD51 filament installation and homology search, and whether such a role depends on specific interplays with BRCA2.

To shed some light on the putative roles of RAD52 during RAD51 filament formation, we have reconstituted in this study the early steps of human HR with purified proteins in a reaction using a synthetic long DNA overhang substrate mimicking the DNA substrates that result after resection of a DSB. We have observed using transmission electron microscopy (TEM) the sequential recruitment to the DNA substrate of RPA, RAD52, and BRCA2, as well as RAD51, in order to understand better the interplay between these different actors in the assembly of the RAD51 filament, its architecture, and its activity. TEM allowed us to directly observe and characterize the molecular features of the transient DNA–protein intermediates generated at different time points during the nucleofilament formation and homologous pairing reactions. Although our results confirm that RAD52 lacks a mediator activity, despite its ability to tightly bind RPA-coated ssDNA, we observed it could inhibit (and modulate) the mediator role of BRCA2 and form shorter RAD52- and RAD51-containing mixed filaments that are more flexible and might thus sustain the more efficient homology search and formation of synaptic complexes and D-loops, permitting as well more frequent multi-invasion events. In line with our in vitro observations, we also confirmed the existence in vivo of RAD51 and RAD52 assemblies on the same substrate after the induction of DSBs.

## Results

### RAD52 is involved in HR in response to DSBs

Different from its yeast homolog where Rad52 defines the epistasis group of proteins for HR pathways (63), mammalian RAD52 is not an essential protein for HR and *RAD52* –/– mice are viable and fertile and only show a slight decrease in HR activity (60). In general, deficiency of RAD52 is not linked to direct evidence of damage sensitivity, but the fact that the overexpression of *RAD52* in mammalian cells improves their resistance to ionizing radiation suggests a role in response to DNA damage by HR (64). To further inquire its role in the HR pathway, we decided to use two different dedicated reporter systems in parallel, the DR-GFP (65) and the CRISPR-LMNA homology-directed repair (HDR) assays, which are able to monitor mild recombination phenotypes (Fig 1). The DR-GFP

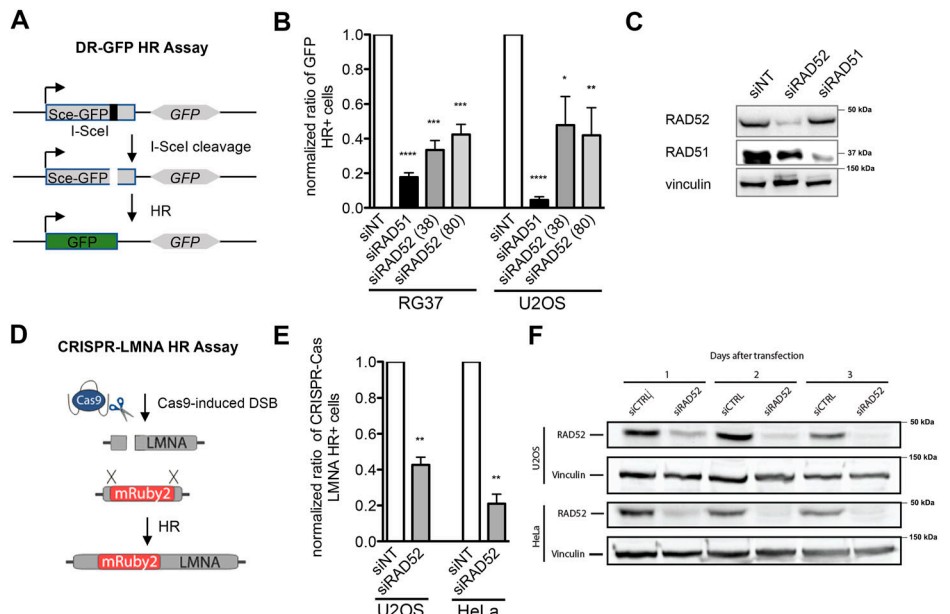

**Figure 1. Homologous recombination is affected in cells silenced for RAD52.**
**(A)** Schema of the DR-GFP reporter assay. Sce-GFP is a modified GFP gene containing an I-SceI site and in-frame termination codon. *GFP* is an 812-bp internal GFP fragment. Repair by HR results in the restoration and expression of a functional GFP gene rendering cells fluorescent. **(B)** Normalized frequency of HR in RG37 and U2OS cell line knockdown for RAD51 or RAD52 (with two distinct siRNAs). Bars are the mean ± SEM and reflect the results of six independent experiments. **(C)** Western blot showing the depletion of RAD51 and RAD52 in cells transfected with RAD51 and RAD52 siRNAs. **(D)** Schema of the CRISPR-LMNA HR assay. Cleavage of the LMNA gene using Cas9- and LMNA-targeting gRNA results in homology-directed repair using a co-transfected donor template that places the coding sequence for the Ruby fluorescent protein in-frame with exon 1 of LMNA, resulting in nuclei with fluorescent nuclear lamina (fluorescent micrograph). **(E)** Amount of Ruby2+ cells (HR+ cells) among GFP+ cells (transfected cells) is represented for U2OS and HeLa cells on the graphs as a ratio normalized to a control. Three independent experiments were conducted in each cell line, and at least 500 GFP+ cells were analyzed in each condition and in each replicate. **(F)** Western blot showing the depletion of RAD52 in cells transfected with RAD52 siRNA (**** = $P < 0.0001$, *** = $P < 0.001$, ** = $P < 0.01$, and * = $P < 0.05$, unpaired $t$ tests, two-tailed).

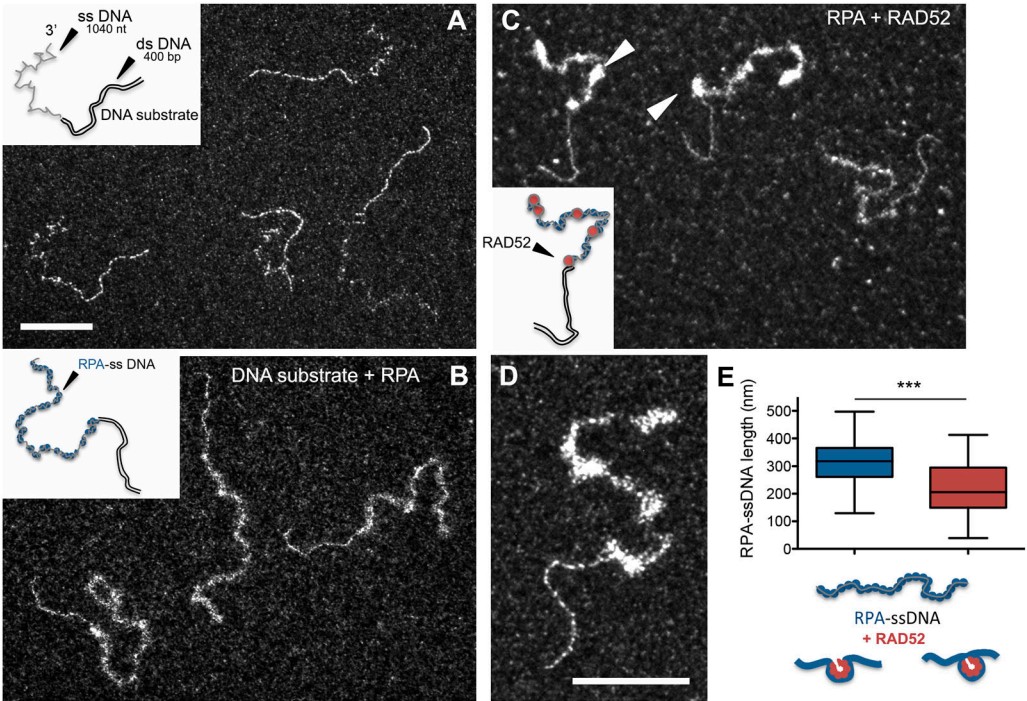

**Figure 2. RAD52 binds to RPA-ssDNA.**
**(A, B, C, D)** Representative TEM images of the DNA–protein complexes in the reactions with insets of schematic drawings of the molecules. **(A)** DNA substrate containing a 400-bp dsDNA extended with a 1,040-nt-long ssDNA overhang. **(B)** Saturated concentration of RPA protein (0.45 $\mu$M) is incubated with 15 $\mu$M DNA substrate and covers the ssDNA part of the substrate. **(C)** Then, 0,25 $\mu$M RAD52 is added to the reaction, and discrete complexes are detected and pointed with arrows. **(A, B, C, D)** Same magnification (the scale bar represents 100 nm). **(D)** Zoom on a RPA-RAD52-DNA complex (the scale bar represents 100 nm). **(E)** Measurements of the length of RPA-ssDNA in the absence (in blue) or in the presence (in red) of RAD52. RAD52 induces a 30% length reduction. The cartoon shows how RPA-ssDNA or free ssDNA could be wrapped around RAD52 oligomer and how it could explain the shortening of the complex. Bars are the mean ± SEM and reflect the results of two independent experiments (*** = $P < 0.001$, unpaired $t$ tests, two-tailed).

assay tracks the restoration of GFP expression as a result of an I-SceI–induced (DSB) gene conversion (Fig 1A), and cellular green fluorescence is then assessed to quantify HDR frequency (Fig 1B). The CRISPR-LMNA HDR assay uses Cas9 cleavage of the LMNA gene, which, if repaired by HDR, results in the expression of the LMNA-Ruby fusion protein (Fig 1D). As a result, cells display a Ruby2+ fluorescent nucleus and the amount of Ruby2+ cells (HR+ cells) among GFP+ cells (transfected cells) is quantified (Fig 1E). Although the silencing of recombinase RAD51 through siRNA quasi-totally abolished HR gene conversion in the DR-GFP system, the *RAD52* knockdown (Fig 1C and F) only led to a partial decrease in HDR that dropped from 50% to 80% of its WT levels in RG37, U2OS, and HeLa cell lines in both HDR assays.

## RAD52 binds and compacts RPA-covered ssDNA

To further clarify the putative role of RAD52 in the RAD51 nucleofilament formation and its behavior on resected ends, we implemented biochemical reactions to test its ability to bind RPA-covered ssDNA on an ss-dsDNA hybrid substrate mimicking the ssDNA overhang generated after a resected DSB end (Figs 2 and S1C and D). The substrate contained a dsDNA region of 400 bp extended with a 3′ overhang of 1,040 nucleotides to mimic the structure and approximate length found in DSBs processed in vivo (Fig 2A and Table 1) (67). After incubation of this 3′-overhang substrate with saturating concentrations of RPA, we observed its binding, covering the ssDNA segment of the substrate (Fig 2B). Naked ssDNA segments fold stochastically as can be observed in Fig 2A. After the addition of RPA, which helps destabilize DNA secondary structures, these ssDNA segments were deployed at the surface of the TEM grid (Fig 2B compared with Fig 2A). The dsDNA part of the substrate remained linear and well spread on the grid surface. The addition of RAD52 to the reactions containing RPA first revealed that RAD52 can bind to RPA-ssDNA to form discrete complexes (Fig 2C and D) accompanied by a significant decrease in the RPA-ssDNA complex length of nearly 30% of its size before RAD52 addition (from mean length 314 ± 77 nm to 220 ± 93 nm; Fig 2E). RAD52 is known to oligomerize adopting a ring (or open ring) shape (50, 51, 52, 68); such oligomerization could explain this shortening as the RPA-ssDNA fibers fold or wind around RAD52 oligomers. By estimating an average number of 4,5 RAD52 oligomers per RPA-ssDNA overhang and taking into account the measured length of RAD52-RPA-ssDNA compared with RPA-ssDNA (in nm) relative to the 1,040-nucleotide-long ssDNA, we estimate that one RAD52 oligomer would bind to ~69-nucleotide ssDNA. Although we did not observe any interaction of RAD52 with the dsDNA segment of the substrate, we noticed the very frequent localization of RAD52 complexes at the dsDNA-ssDNA junction (37% ± 7%) (Fig S2A–D). RAD52 could also interact tightly with the naked ssDNA of our DNA substrate, but in these conditions, it did not form discrete complexes as those observed in the presence of RPA, but instead, it showed a tendency to form aggregates and DNA intramolecular bridges (Fig S2E and F), as previously observed (69). The presence of a four-nucleotide-long overhang was sufficient to promote RAD52 recruitment at the ss-dsDNA junction. Using these short overhangs, we could observe either discrete RAD52-DNA complexes or end-to-end DNA aggregates (Fig S2G–I). This aggregation of DNA molecules was also obtained by increasing the RAD52 concentration, revealing an

**Table 1.   Sequences of primers and siRNAs used in the study.**

| Primers | |
|---|---|
| Cy5-2574+ | 5′-CGACGCTCAAGTCAGAGG-3′ |
| Biotin-4014- | 5′-GGATCTCAACAGCGGTAA-3′ |
| Biotin-2574+ | 5′-CGACGCTCAAGTCAGAGG-3′ |
| 2976- | 5′-ATTTTTGTGATGCTCGTC-3′ |
| siRNA DR-GFP assay | |
| Control siRNA | 5′- AUGAACGUGAAUUGCUCAA-3′ |
| siRNA RAD51 | 5′-GUGCUGCAGCCUAAUGAGA-3′ |
| siRNA RAD52 (38) | 5′-CCAACGCACAACAGGAAAC-3′ |
| siRNA RAD52 (66) | 5′-GGUCAUCGGGUAAUUAAUC-3′ |
| siRNA CRISPR-LMNA assay | |
| Control siRNA | 5′-UUCGAACGUGUCACGUCAA-3′ |
| siRNA RAD52 | 5′-CCAACGCACAACAGGAAAC-3′ |
| siRNA PLA | |
| Control siRNA (SMART pool) | 5′-UGGUUUACAUGUCGACUAA-3′ |
| | 5′-UGGUUUACAUGUUGUGUGA-3′ |
| | 5′-UGGUUUACAUGUUUUCUGA-3′ |
| | 5′-UGGUUUACAUGUUUUCCUA-3′ |
| siRNA RAD51(SMART pool) | 5′-UAUCAUCGCCCAUGCAUCA-3′ |
| | 5′-CUAAUCAGGUGGUAGCUCA-3′ |
| | 5′-GCAGUGAUGUCCUGGAUAA-3′ |
| | 5′-CCAACGAUGUGAAGAAAUU-3′ |
| siRNA RAD52 | 5′-CCAACGCACAACAGGAAAC-3′ |

intrinsic property of RAD52 to gather and hold together ssDNA molecules, maybe in relation to its ability to anneal complementary DNA sequences. These properties of RAD52 can be explained by the presence of two DNA-binding sites in the RAD52 oligomeric ring, a primary ssDNA-binding site along the outer circumference of the ring, and a second DNA-binding site able to bind either ssDNA or dsDNA (50, 52, 68, 70, 71).

## RAD52 bound to RPA-ssDNA inhibits the BRCA2 mediator activity

We then decided to test the putative mediator activity of RAD52 and BRCA2 individually or combined by analyzing their ability to recruit RAD51 to the RPA-covered 3′-overhang DNA substrate, thus allowing the RAD51 filament formation (Figs 3 and S1A–D). Precisely, the DNA substrate was first incubated with saturating amounts of RPA to generate RPA-covered ssDNA overhangs, and RAD51 was then added in the presence of either purified RAD52 or BRCA2 proteins. The presence of prebound RPA clearly inhibited the RAD51 loading and filament assembly on the ssDNA segment of the DNA substrate (Fig 3A), which was consistent with the fact that in vitro formation of RAD51 filaments on ssDNA usually requires RPA to be added to the reaction after RAD51 in the absence of mediators. Interestingly, in these conditions, RAD51 was able to polymerize on the dsDNA section of the substrate forming continuous filaments, the ssDNA still being covered by RPA (Fig 3A and E). It is worth noting that in the

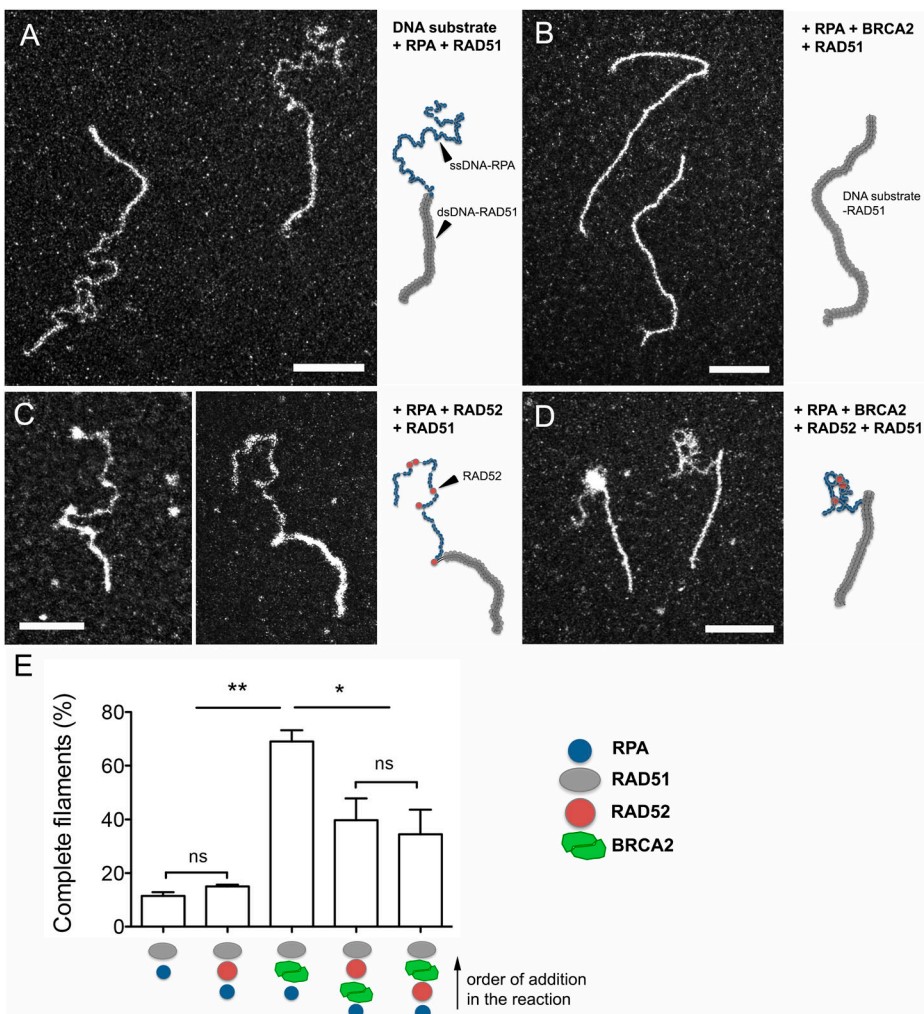

**Figure 3. Early introduction of RAD52 inhibits the RAD51 assembly on ssDNA.**
**(A, B, C, D)**. Representative TEM images of the DNA–protein complexes in the reactions with insets of schematic drawings of the molecules. **(A)** 15 µM DNA substrate is first incubated with a saturated concentration of RPA (0.45 µM), and then, RAD51 (5 µM, one protein per three nucleotides) is added to the reaction. RAD51 filament assembles on the dsDNA part of the substrate without replacing RPA on the ssDNA part. **(B)** Substoichiometric amount of BRCA2 (2 nM) is introduced in the reaction simultaneously with RAD51 and promotes the formation of complete filaments (on dsDNA but also ssDNA parts of the substrate)by replacing RPA along ssDNA, highlighting the BRCA2 mediator role. **(C)** 0.25 µM RAD52 was added to the reaction simultaneously with RAD51. RAD52 binds to RPA-ssDNA but does not allow RAD51 filament installation on ssDNA, confirming that human RAD52 does not exhibit any mediator activity as BRCA2. **(D)** RAD52 and BRCA2 are introduced together in the reaction. RAD52 binding to RPA-ssDNA predominantly prevents the BRCA2-mediated nucleation of RAD51 on ssDNA and subsequent complete filament formation, thus partially inhibiting the BRCA2 mediator activity of BRCA2. All scale bars represent 200 nm. **(E)** Quantification of complete RAD51 filaments (assembled on dsDNA and ssDNA parts of the DNA substrate) in the DNA–protein samples. Bars are the mean ± SEM and reflect the results of three independent experiments (** = $P < 0.01$ and * = $P < 0.1$, unpaired $t$ tests, two-tailed).

absence of RPA, we found that human RAD51 similarly binds to ssDNA or dsDNA (Fig S3A), in contrast to *S. cerevisiae*'s Rad51 or *Escherichia coli*'s RecA recombinases, which exhibit a preferential affinity for ssDNA (72). This specific property of hRAD51 raises the question of the function of potential transitory binding of RAD51 to dsDNA, which is negatively regulated, as observed in some contexts in vivo (14).

The introduction of BRCA2 at substoichiometric concentration (from 1 to 5 nM—one protein for one DNA substrate) promoted the loading and assembly of RAD51 by replacing RPA on the ssDNA part of the overhang to form complete filaments on the whole substrate (Fig 3B). This bona fide mediator activity was illustrated by the presence of 69% complete RAD51 filaments in the presence of 1 nM BRCA2 compared with 10% in its absence (Fig 3E). This mediator activity of BRCA2 could not be substituted by adding RAD52 to the reaction (at concentrations ranging from 0.1 to 1 µM). We could not observe RAD51 loading on RPA-covered ssDNA after RAD52 addition in any of the conditions tested (Fig 3C and E). Again, we detected RAD52 bound at discrete positions on RPA-ssDNA, this binding being coupled to a significant reduction in the RPA-ssDNA complex length. When both BRCA2 and RAD52 were introduced together in

the reaction, regardless of the order they were added, partial inhibition of the BRCA2 mediator activity was observed with roughly a 50% decrease of the complete filaments that could be observed (Fig 3D and E). This reduction illustrates a competition between RAD52 and RAD51-BRCA2 for binding to RPA-ssDNA.

## RAD52 participates in the formation of mixed, fragmented, and flexible filaments, whereas BRCA2 catalyzes the formation of long and continuous RAD51 filaments

To further analyze the effect of RAD52 and/or BRCA2 on the architecture and activity of RAD51 filaments, we carried out a series of reactions where we first incubated the DNA overhang substrate with RAD51 and either RAD52 or BRCA2, followed by the addition of a non-saturating amount of RPA to help in the filament installation through the removal of ssDNA secondary structures upon RAD51 polymerization (Fig 4).

The presence of BRCA2 (2 nM) in the reaction promoted the formation of long, complete, and continuous filaments. These filaments showed a regular helical architecture without interruptions when observed by negative staining TEM (Fig 4A, B, and H). Despite

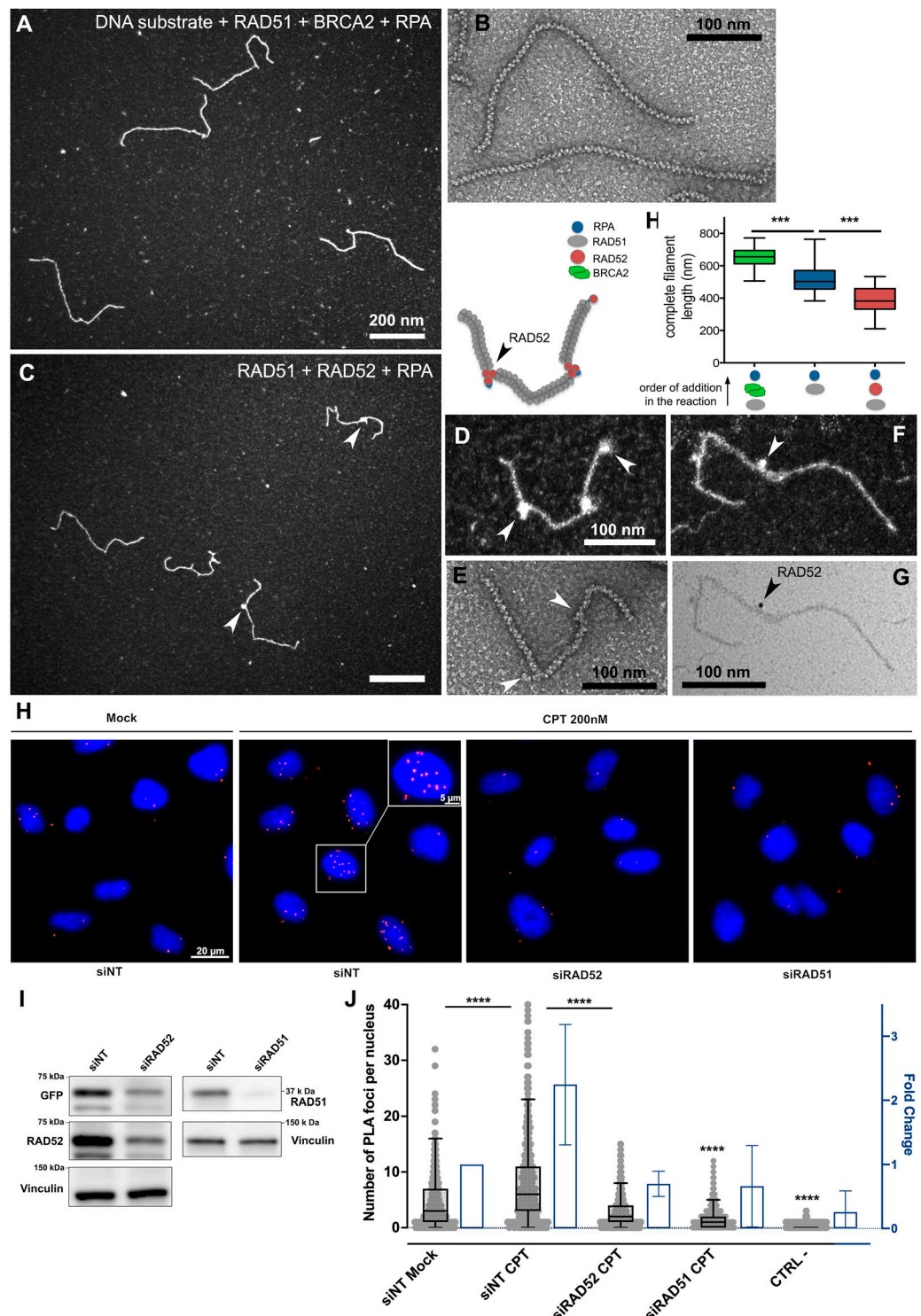

**Figure 4. BRCA2 induces the formation of long and continuous RAD51 filaments, whereas RAD52 forms with RAD51 mixed and fragmented filaments, and RAD51 and RAD52 are co-recruited in vivo in CPT-treated cells.**
**(A)** Representative TEM image of RAD51 filaments formed in the presence of BRCA2. Briefly, 15 $\mu$M DNA substrate was incubated with 5 $\mu$M RAD51 and 2 nM BRCA2, for 3 min at 37°C, and then, 0.15 $\mu$M RPA (a subsaturated concentration) was added to the reaction and incubated for 15 min more. **(B)** Negative staining TEM image showing the same long and continuous filaments. **(A, C)** Mixed filaments formed in the same conditions as (A) but in the presence of 0.2 $\mu$M RAD52 instead of BRCA2. Arrows point bright spots in the filaments, suggesting RAD52 discrete complexes bound to the filament. **(D)** Zoom on a mixed filament with a schematic drawing of the molecule above. **(E)** Negative staining image of a mixed filament. **(F, G)** Darkfield and brightfield TEM imaging of immunostaining experiment of the mixed RAD51-RAD52 filament specifically showing the presence and localization of RAD52 using anti-RAD52 antibody and gold bead–coupled secondary antibody. The arrow points to the gold bead (with an increased electron density). **(H)** Measurement of the length of complete filaments formed in the presence of BRCA2 (in green); pure filaments formed by RAD51, then low

the fact that BRCA2 is a 380-kD protein forming a dimer in solution, and thus its size would allow its detection within the RAD51 filament (73), we could not detect any BRCA2 binding onto RAD51 filaments. The inability to detect BRCA2-bound intermediates strongly suggests that BRCA2 mediates RAD51 filament formation without remaining stably bound to the DNA in our experimental conditions. RAD51 filaments formed in the presence of BRCA2 in these conditions were 27% longer than those formed in its absence (648 ± 66 nm, which corresponds, after correction for DNA extension by RAD51, to 1270 ± 129 pb; Fig 4H), indicating that BRCA2 had a positive effect not only on RAD51 nucleation but also on the filament elongation and stability. Moreover, the addition of BRCA2 on pre-assembled filaments also resulted in a significant increase in their length, further confirming a role of BRCA2 in filament stability. Different from what was observed in the presence of BRCA2, RAD51 filaments formed in the presence of RAD52 (0.25 $\mu$M) displayed a number of discontinuities and the presence of discrete complexes in the form of two to three bright spots per filament, detectable using either a positive or a negative staining of the sample (Fig 4C–E). The presence of these clusters or complexes suggested RAD52 was part of the RAD51 filament and its helicity interruption was often associated with a "kink" in the filament. To specifically show and localize RAD52 inside the filaments, we performed immunolabeling assays using an anti-RAD52 antibody and a secondary antibody coupled to gold beads. This labeling enabled observing, indeed, how mixed RAD51-RAD52 filaments consist of tracts of the RAD51 filament interrupted by RAD52 oligomers, specifically tagged by the presence of gold beads (Fig 4F and G). Again, RAD52 binding to the RAD51 filament was associated with a significant 25% reduction in the filament length, in line with the putative winding of the ssDNA or RPA-ssDNA fiber around the RAD52 oligomers (Fig 4H). To determine whether the presence of RAD52 within the filaments induced a local change in their flexibility or bending, we measured the $\theta$ angles formed at the filament bends (change in the direction of the filament) and whether they are associated with the presence of RAD52 (Fig S3D–F). Interestingly, we observed a significant increase in the number of kinks in the filaments and an increase in the $\theta$ angle from 63.5° ± 49 to 90° ± 69, of which most of the large angles are associated with the presence of RAD52 (Fig S3E and F, red dots on the graph), showing that RAD52 induced local changes in RAD51 filament flexibility allowing changes in the filament direction. Increasing the RAD52 concentration above 1 $\mu$M, RAD52 did not induce an increase in the number of discrete spots per filament but rather caused a background covering with particles (Fig S3B) pointing to competition between RAD51 and RAD52 for ssDNA binding, thus allowing only a limited number of oligomers to interact with the filament. The addition of RAD52 in a later time point of the reaction, over an already preformed RAD51 filament, did not change the shape, length, and helical architecture of the filament, indicating that RAD52 in this condition was excluded from the filament, a result in agreement with reference 74.

## RAD52 and RAD51 interact in situ after DSB induction

If RAD51 and RAD52 form mixed filaments in vitro, we suspected them to also coexist in the same nucleofilament in the cell during DNA repair by HR. To test this hypothesis, we decided to use the RAD51-RAD52 proximity ligation assay (PLA) in U2OS cells treated with camptothecin (CPT). The PLA allows amplifying a fluorescent signal when the two protein targets are immunolabeled with a proximity of less than 40 nm. After browsing several commercial anti-RAD52 antibodies that were poorly selective, we decided to use an epitope tag version of RAD52 (U2OS RAD52-GFP cells) and the much more selective anti-GFP antibody. PLA in such settings enabled us to detect a substantial proximity of RAD51 and RAD52-GFP, 5 h after a 200 nM camptothecin (CPT) treatment, known to induce DNA damages, including DSBs. We confirmed that the number of PLA signals in the form of discrete foci was dependent on RAD51 and RAD52 because their silencing significantly decreased the PLA from 7.9 ± 7.6 to 1.7 ± 1.9 and 2.7 ± 2.7 foci, respectively (Fig 4I and J). Our results, overall, demonstrate that RAD51 and RAD52 are co-recruited into chromatin and interact in situ after DSB induction by CPT treatment.

## Human RAD51 displays an important ability to contact a dsDNA donor independently of the presence of sequence homology

It was previously shown that human RAD51 can form D-loops on its own, but only in the presence of calcium, this activity being stimulated by RAD54, whereas its yeast homolog absolutely requires Rad54 to form synaptic complexes and D-loops (75, 76, 77, 78). We decided to test whether this strand invasion activity of RAD51 is influenced by the presence of BRCA2 or RAD52 (Fig 5).

A RAD51 filament was first assembled on the DNA substrate, followed by the addition of a dsDNA donor (containing homologous or heterologous sequences) to the reaction (Fig 5A). Homology search and strand invasion processes are characterized by the formation of joint molecules where the presynaptic filament pairs with the homologous dsDNA donor (78). In our experimental approach, half the sample was subjected to TEM analysis, whereas the other half was analyzed in a traditional D-loop assay involving deproteinization of the reaction products, DNA species separation on a gel, and quantification (Fig 5A). The D-loop is defined as the joint-molecule product of the incorporation of the invading strand into a homologous dsDNA donor, resulting in the disruption of its original base pairing and replacement by a newly formed heteroduplex. In the D-loop assay, D-loop products are characterized by the intertwining of the invading strand with its complement in the donor; they are stable in the absence of proteins and then resistant to deproteinization. With our overhang substrate, in the presence of RAD51 and a homologous dsDNA donor, 4.9% D-loops were observed at 20 min after the dsDNA donor addition in the

mount of RPA (in blue); and mixed RAD51-RAD52 filaments (in red). Bars are the mean ± SEM and reflect the results of two independent experiments (*** = $P$ < 0.001, unpaired Mann–Whitney, two-tailed). **(I)** Western blot analysis showing the depletion of RAD51 and RAD52 in U2OS cells transfected with RAD51 and RAD52 siRNAs. **(J)** RAD51-RAD52 proximity ligation assay in the U2OS RAD52-GFP cells treated or not with camptothecin (CPT) showing that RAD51 and RAD52 are co-recruited into chromatin after DSB induction (with a proximity of less than 40 nm). The graph represents the number of foci per nucleus and the fold change compared with the non-treated cells. A small amount (less than 1%) of cells with a very high number of foci was excluded from the quantification (**** = $P$ < 0.0001, unpaired Mann–Whitney, two-tailed).

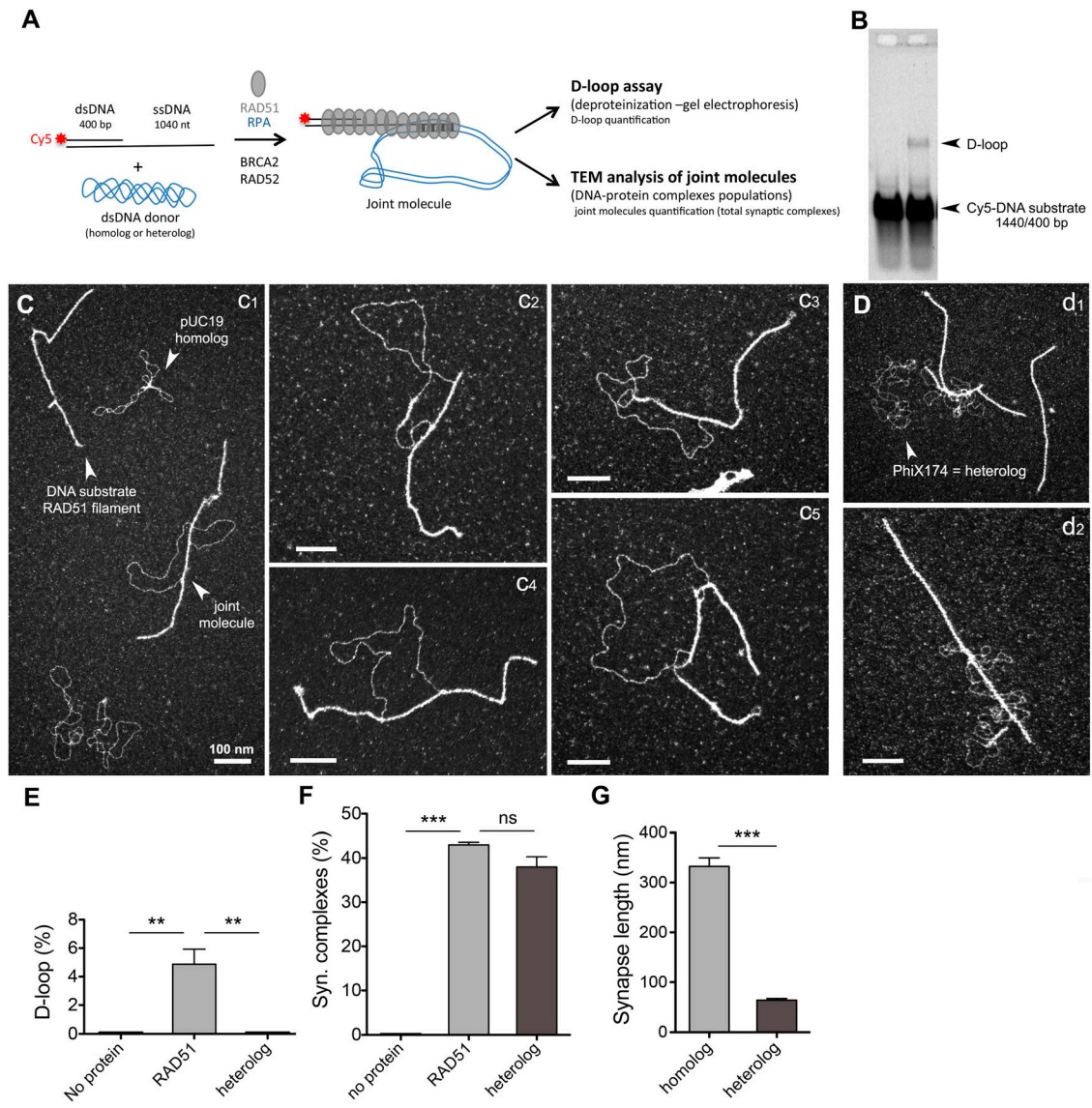

**Figure 5. Human RAD51 filament is highly active in contacting the dsDNA donor to form synaptic complexes.**
**(A)** Schema of the D-loop in vitro reaction. The RAD51 filament is preformed on the ss-dsDNA substrate, and then, 25 nM in molecules of a dsDNA heterologous or homologous donor is added to the reaction. Concretely, 15 $\mu$M DNA substrate was incubated with 5 $\mu$M RAD51, for 3 min at 37°C, then 0.15 $\mu$M RPA (a subsaturated concentration) was added to the reaction and incubated for 15 min more, and finally, 25 nM in molecules of a dsDNA heterologous or homologous donor was added to the reaction. Three independent reactions have been performed. For each reaction, one part is deproteinized and then run on a 1% agarose gel. D-loops are quantified using ImageJ software. The second part is diluted, spread on a microscopy grid, and analyzed. **(B)** Representative gel of the D-loop assay. **(C)** (1, 2, 3, 4, 5) TEM images of the D-loop reaction performed using a homologous dsDNA donor. **(D)** (1, 2) TEM images of the D-loop reaction performed using a heterologous dsDNA donor. The scale bars in all images represent 100 nm. **(E)** Quantification of the D-loop yield (for three independent D-loop reactions) (** = $P < 0.01$, unpaired $t$ tests, two-tailed). **(F)** Quantification of synaptic complexes formed by the pairing of the RAD51 filament with the dsDNA donor. The synaptic complex percentage represents the number of filaments paired with the dsDNA donor by the total number of filaments (paired and non-paired) (*** = $P < 0.001$, unpaired $t$ tests, two-tailed; ns = non-significant). **(G)** Measurement of the synapse length (in nm).
Source data are available for this figure.

reaction (Fig 5B and E). Joint molecules were also directly visualized using TEM (Fig 5C and D). TEM is ideal for directly observing different populations of DNA and protein: DNA complexes. In the case of the D-loop reaction, this allowed us to study the intermediates that precede strand intertwining, specifically protein-mediated pairings such as synaptic complexes where the invading DNA is not intertwined with the donor molecule, these joint molecules being susceptible to deproteinization. We clearly distinguished joint-molecule synaptic complexes resulting from the interaction between the nucleoprotein filament and the duplex DNA donor, coexisting with the reaction substrates (free RAD51 filaments on the 3' substrate and supercoiled DNA; Fig 5C). Surprisingly, at 20 min after dsDNA donor addition, 43% of synaptic intermediates were counted, revealing the extraordinary capacity of the human RAD51 filament to establish stable contacts with dsDNA (Fig 5F). Only 9% of synaptic contacts resulted in the formation of D-loops by complementary sequence

alignment (resistant to deproteinization and quantified by gel), indicating that the vast majority of synaptic complexes observed by TEM are potentially three-stranded intermediates maintained by RAD51. Observation by TEM also enables the characterization of the joint-molecule architecture, as we can precisely determine where the proteins are bound and thus measure the DNA length, as well as the synaptic part of the joint molecules. Analyzing RAD51-mediated synaptic complexes (SCs), we observed that the contact zone (synapse) between the filament and the dsDNA donor remained covered by RAD51 with an average contact length of 332 ± 60 nm, equivalent to 651 bp when corrected for the extension by RAD51 (Fig 5G). The topology of the negatively super-coiled dsDNA donor was modified and relaxed as a consequence of the SC and D-loop formation (3, 28, 79, 80). Interestingly, when the added dsDNA donor was heterologous, 38% of SCs were observed (Fig 5D, F, and G). Thus, the formation of the major part of these stable RAD51-mediated contacts was independent of the presence of homology. However, qualitatively, joint molecules formed with heterologous dsDNA showed distinct features in comparison with the homologously paired SCs. These nucleo-protein filament interactions with heterologous DNA were mainly characterized by short contacts (<50 nm), and the dsDNA did not become topologically relaxed.

### RAD52 promotes synaptic complexes, D-loop formation, and multi-invasions

As demonstrated above, RAD52 forms with RAD51 mixed and segmented filaments. We further tested the activity of these fil-aments to pair with dsDNA and form SCs and D-loops (Fig 6). Strikingly, we observed that the presence of RAD52 rendered filaments 1.7-fold more active in contacting dsDNA homologous donors, with a great increase in SC formation (from 43% to 72% when 200 nM RAD52 was added to the reaction), demonstrating that mixed filaments are more efficient in establishing contacts with the dsDNA donor (Fig 6A and C). This positive effect on SC formation was associated with a significant increase in stable D-loop intermediate formation from 4.9% to 13.8% (in the absence and the presence of RAD52, respectively, Fig 6D and E). When RAD52 is titrated into the D-loop reaction, D-loop products reach a peak at 250 nM RAD52 (Fig S4A; the optimal D-loop yield was 18.4% for 250 nM), whereas RAD52 does not form D-loop on its own (Fig S4C). This positive effect of RAD52 was partially independent of homology because the proportion of joint molecules formed between the mixed RAD51-RAD52 filament and a dsDNA heter-ologous donor was also significantly enhanced in comparison with pairings involving pure RAD51 filaments and heterologous donors (Fig 6C). Increasing the RAD52 concentration in the re-action did not increase the representativeness of RAD52 within the filaments (as stated above) nor the quantity of SCs and D-loops but rather led to the aggregation of dsDNA molecules with filaments, highlighting the high potential of RAD52 oligomer to interact and bridge multiple DNA molecules, thus generating aggregates. In contrast, the addition of BRCA2 to the reaction had no significant effect on the proportion of joint molecules quan-tified either by TEM or in the D-loop assay, although a slight (but not significant) increase in D-loop intermediates was reproducibly observed, which may be explained by the fact that filaments formed in the presence of BRCA2 are longer and more stable (Figs 6C–E and S4B). Considering the architecture of SCs formed by pairing mixed RAD51-RAD52 filaments with a homologous dsDNA donor, we did not show any change in the synaptic contact length. However, we noticed the frequent localization of RAD52 com-plexes on both sides of the junction zone, as if the presence of RAD52 was delineating the boundaries of the synapse. RAD52 was not detected along the junction zone. This result was confirmed by the specific labeling of RAD52 using an immunodetection assay (Fig 6A, B, and F). RAD52 could indeed play a role in delineating and then restricting the synaptic contact zone. Compatible with this scenario, a first oligomer of RAD52 may initiate contact with the donor dsDNA and lead to an extension of the synaptic area to the next oligomer when homology is found.

Finally, we identified an interesting elevated rate of multi-invasion events within the population of RAD51-RAD52–mediated joint molecules involving contacts with two or three donor DNAs (Fig 6F and G). These multi-invasions amounted to 35% of the events observed in the presence of RAD52, compared with the 7% of these multi-invasions in the control conditions in the absence of RAD52. Again, bright discrete complexes were detected at the edges of the synaptic zone in these multi-invasion events. This result suggests that mixed RAD51-RAD52 filaments are active not only in their ability to contact dsDNA and form joint molecules but also in establishing multiple synapses by contacting and holding several homolog DNAs at once. Our results thus point to a critical role of RAD52 in the proposed alternative multi-invasion–induced rearrangement mechanism described by reference 66 (see the Discussion section).

## Discussion

The participation and specific role of human RAD52 in the HR process, more precisely its cooperation with RAD51, remain con-troversial. In this study, using a combination of biochemistry and high-resolution TEM imaging, we tested the putative collaboration of both BRCA2 and RAD52 in the substitution of RPA-coated ssDNA to give rise to the nucleation and growth of a RAD51 nucleofilament. Our analysis confirmed the ability of BRCA2 to promote RAD51 nucleation on ssDNA at a substoichiometric concentration (Fig 3), strongly confirming its role as the key RAD51 mediator in humans. We were also able to show how RAD52 tightly interacts and is able to lead RPA-coated ssDNA into a higher degree of compaction (Fig 2), thus partially inhibiting or limiting the RAD51 nucleation on the ssDNA section of the resected DNA ends (Fig 3C and D). This result demonstrates that RAD52 does not show a mediator activity in our conditions but helps limiting the length and continuity of the nucleofilaments by inhibiting RAD51 loading on RPA-ssDNA. Our data highlight the existence of two types of nucleofilaments depending on the presence and stoichiometry of these RAD51 partners: long, regular, and continuous filaments mediated by an unopposed BRCA2 activity; and shorter, discontinuous filaments occurring in the presence of RAD52, interspersed by clusters of RAD52 likely in the form of oligomers, which we have defined as

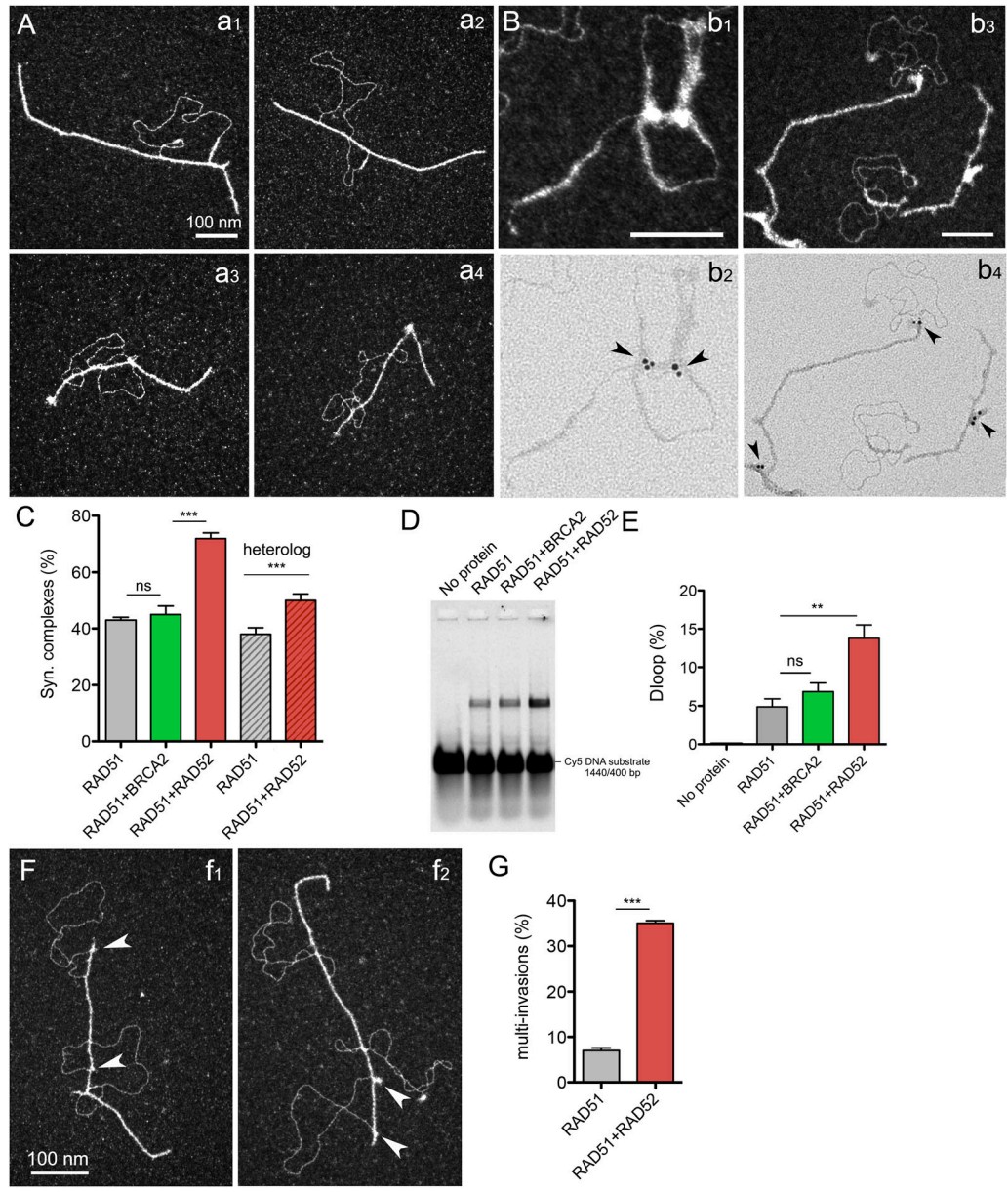

**Figure 6. Mixed filaments formed in the presence of RAD52 are more active in contacting dsDNA and forming synaptic complexes, D-loops, and multi-invasions.**
**(A)** TEM images of joint molecules formed in the D-loop reaction by the pairing of mixed RAD51-RAD52 filaments with a homologous dsDNA donor. 15 $\mu$M DNA substrate was incubated with 5 $\mu$M RAD51 and 0.2 $\mu$M RAD52, for 3 min at 37°C, then 0.15 $\mu$M RPA (a subsaturated concentration) was added to the reaction and incubated for 15 min more, and finally, 25 nM in molecules of a dsDNA heterologous or homologous donor was added to the reaction during 20 min at 37°C. **(B)** Immunolabeling of the previous reaction using specific anti-RAD52 antibodies. **(C)** Quantification of synaptic complexes formed during the D-loop reaction in the presence of 2 nM BRCA2 or 0.2 $\mu$M RAD52 and also in the presence of non-homologous (heterologous) dsDNA (*** = $P < 0.001$, unpaired $t$ tests, two-tailed; ns = non-significant). **(D, E)** Deproteinized D-loop assay run on a 1% agarose gel and quantification of the D-loop yield (E) (** = $P < 0.01$, unpaired $t$ tests, two-tailed; ns = non-significant). **(F, G)** TEM images of multi-invasions identified in the D-loop reaction in the presence of RAD52 and their quantification (G) (*** = $P < 0.001$, unpaired $t$ tests, two-tailed).

mixed RAD51-RAD52 filaments (Fig 4). We were able to characterize these mixed filaments as more flexible and more prone to establish contacts with a dsDNA donor thus creating synaptic intermediates and D-loops (Fig 6). According to these properties, we observed how these mixed filaments are more proficient in establishing simultaneous multi-invasions of segmented RAD51 filaments in vitro (Fig 6F and G). In cells, these molecular observations are backed by the detection of a close-range proximity of RAD51 and RAD52 in PLAs,

indicative of interaction in situ 5 h after DSB induction, compatible with the existence of these mixed filaments in vivo. In light of these results, we propose a model in which the two partner proteins could act sequentially: (1) firstly, individually, by protecting the newly formed ssDNA, and (2) secondly, synergistically, by promoting an efficient RAD51 nucleation filament growth and HR stimulation with a limited length and more flexible synaptic abilities (see the model in Fig 7).

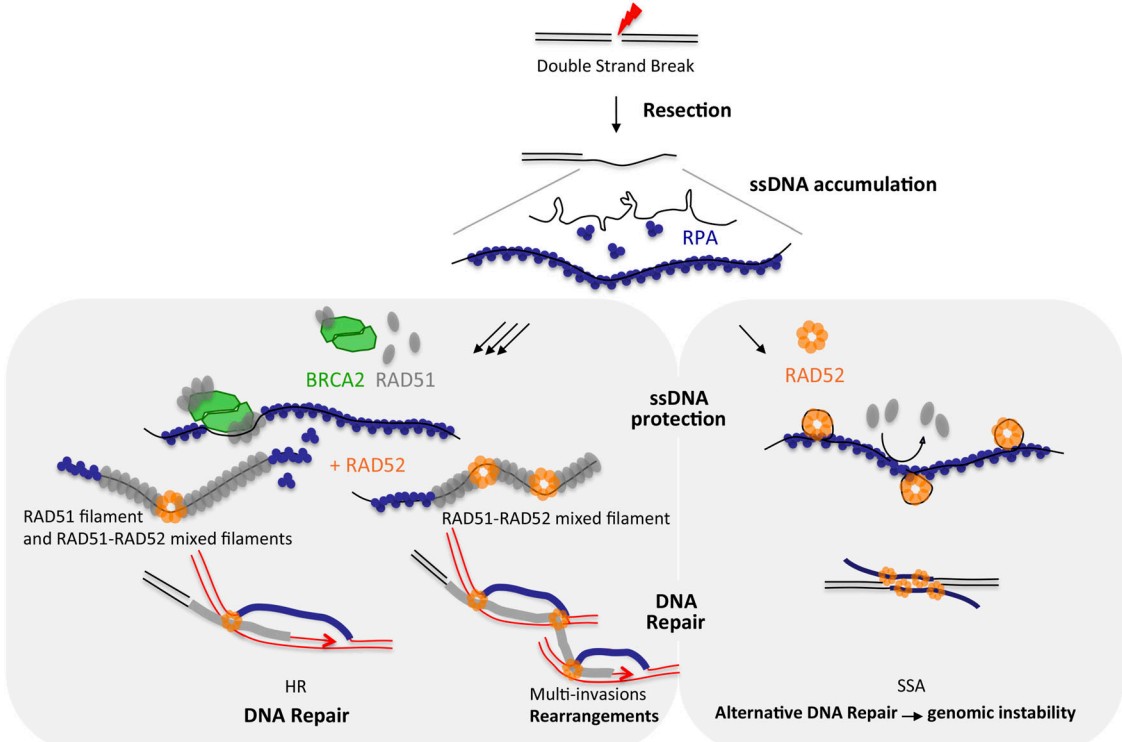

**Figure 7.  Model explaining the interplay between RAD52 and BRCA2 in HR early steps.**
Double-strand break associated with resection leads to the accumulation of long ssDNA (overhang) that is rapidly covered by RPA protein. RAD52 can directly bind and compact RPA-ssDNA, inducing ssDNA protection, and, in some context, associate with genomic instability. On the contrary, BRCA2 can bind and load RAD51 on RPA-ssDNA to promote filament assembly, ssDNA protection, and HR. RAD52 not only can participate in the formation of some mixed and more active filaments to probe dsDNA donor and carry out homology search, but is also more prone to multi-invasions and associated rearrangements.

## A molecular basis for the ssDNA protection roles of RAD52

Protective functions of BRCA2 and RAD52 have been previously described, notably during replication stress (81), and might be one of the essential roles of RAD52 in a BRCA2-defective background. At the molecular level, we show how BRCA2 has an unparalleled ability to promote the recruitment and growth of RAD51 filaments, fully covering DNA. This is not the case for RAD52, and thus, its protective role cannot be related to the formation of RAD51 nucleofilaments as an alternative mediator to BRCA2 but rather to its unique ability to freeze RPA-coated ssDNA into more compact conformation, by wrapping the ssDNA around its oligomeric rings in the presence of RPA. Our resource to TEM imaging of RAD52 bound to the ssDNA overhang substrate clearly allows visualizing the preferential localization at the ss-dsDNA junction, which illustrates how RAD52 could specifically block dsDNA access to nucleases, thereby preventing resection. Our observation, thus, nicely complements previous work describing how perturbed RAD52-ssDNA binding results in extensive nascent strand degradation by MRE11 (81). These properties of RPA-ssDNA compaction and preferential binding at ss-dsDNA junctions can be well transposed to a replication fork context, where RAD52 is proposed to play a gatekeeper role (81).

This gatekeeper role can also be important to direct timely repair to the RAD51-mediated processing or to alternative modes instead. The occupancy of ssDNA overhangs after resection by either RAD51 or RAD52 is a determinant to select the repair pathway of the RAD51-dependent gene conversion (GC) or the alternative SSA pathway, respectively (82). Our in vitro observations of the inhibition of RAD51 loading upon early binding of RAD52 to RPA-ssDNA clearly highlight the molecular basis by which RAD52 binding at ssDNA (present at resected DSBs, gaps, or forks) could exert an early role in directing subsequent steps of repair at the same time as preventing extensive degradation of the exposed dsDNA. In certain contexts, though, the precocious or excessive binding of RAD52 to the ssDNA-RPA overhangs may lead to non-conservative SSA that helps seal a break when complementary ssDNA sequences are revealed after resection in a RAD51-independent manner (83, 84), a situation that is exacerbated in the absence of BRCA1 or BRCA2 (85).

## RAD52 roles at the RAD51 nucleofilament level

The recombinogenic function of BRCA2 and RAD52 is related to their cooperation with RAD51 for the formation and activity of the pre-synaptic nucleofilament. The essential roles of RAD52 in a BRCA2-deficient context might be solely explained by either a protective role of ssDNA discontinuities at postreplication gaps and stalled forks or its ability to direct repair to alternative pathways such as SSA. Our results, thus, are essential in clearly demonstrating a role of RAD52 in the canonical HR pathway itself. There have been contradictory reports in the past showing variable phenotypes (reduced, limited, or significant) in reporter systems to evaluate the

RAD52 activity during HR (60, 61, 62), making RAD52 an auxiliary factor for HR. We clearly showed with two different approaches the RAD52 contribution to overall HR efficiency after a single DSB induction (Fig 1). Although this role is not essential and thus can be considered as accessory, it does have a significant impact on the ability of cells to complete HR. Our molecular insight by TEM brings some context to this accessory role of RAD52 during canonical HR. We have identified that mixed RAD51 nucleofilaments can form with the presence of discrete RAD52 complexes within the filament. We propose that according to their characteristics, these discrete complexes are composed of RAD52 oligomers around which ssDNA is wrapped, thus explaining not only the decrease in the length of these mixed RAD51-RAD52 filaments, but also the increase in their flexibility compared with otherwise "pure" or RAD51-only filaments. The accessory role of RAD52 through these mixed filaments is reminiscent of what we already described in a previous work with the yeast *S. cerevisiae* Rad52 protein, which was able to form mixed filaments with Rad51 that were more resistant to the Srs2 helicase antirecombinase activity (86). The presence of human RAD52 within RAD51 filaments could have the same protective function against antirecombinases yet to be fully identified that may promote filament dismantling in a way similar to that described for Srs2 in the yeast. RAD52 will thus work again as a gatekeeper within the RAD51 filament, by modulating the timely use of these filaments in subsequent steps for recombination. Finally, by coexisting with RAD51, RAD52 could play a role later in the strand exchange process, thus participating in second-strand capture, as previously proposed (87).

### RAD52 stimulation of synaptic contacts of RAD51 filaments

Although phage and bacterial recombinases are able to form synaptic complexes (SCs) and D-loops autonomously, eukaryotic Rad51 relies on other proteins to achieve greater complexity of HR regulation. We have previously shown that *S. cerevisiae* (Sc) Rad51 filaments absolutely require ScRad54 to pair with dsDNA and engage in a SC during homology search (78). ScRad54 plays a second essential function by converting the SC into a D-loop through strand alignment and ScRad51 removal (33, 78). Contrary to what is observed in other models, human RAD51 filaments have a strong ability to pair with dsDNA donors and form synaptic complexes (SCs) by themselves without the involvement of any third partner in vitro (75, 76, 77) (Fig 5). Indeed, an unexpectedly high proportion (43%) of RAD51 filaments appeared as dsDNA-paired, suggesting that human RAD51 functions differently and can perform dsDNA probing to form SCs autonomously. From these filaments, only a few proportions were undergoing strand alignment and displacement of their complementary strand to create a D-loop, a distinction that can only be identified thanks to the use of our molecular imaging approaches (Figs 5 and 6). This efficiency of human RAD51 to probe and establish contacts with dsDNA is mostly independent of the homology of the sequences, and a proportion of the SC (38%) can be established with heterologous donors (Figs 5 and 6). However, an extension of the synaptic contact zone was detected in the SC formed with homology-containing DNA donors, suggesting a change in the synaptic architecture once homology is found. We found the presence of RAD52 increased the number of RAD51

synaptic complexes and also had a positive impact on increasing the number of synaptic complexes that are able to convert into D-loops (Fig 6). We also show that by forming fragmented, mixed filaments, RAD51 and RAD52 are more likely to promote simultaneous contacts of the same RAD51 filament with multiple donors. These in vitro multi-invasions (MI) of distinct homologous dsDNA or MI–synaptic complexes were characterized by the presence of RAD52 at the borders of the synaptic zones, and were not observed in the absence of RAD52. The essential role of RAD52 in promoting and stabilizing such MI–synaptic complexes would support the putative conservation in humans of a mechanism recently described in yeast, where HR mediated simultaneous invasions of two intact donors by a unique broken DNA end generating multi-invasions (MI) and byproducts that further associate with chromosomal translocations (66). This mechanism would be of special importance in the context of highly repetitive genome regions, where multi-invasions may lead to cycles of deletion or expansion of repeated sequences. It could also be responsible for some structural variants (characterized by large chromosomal aberrations) that arise during cancerogenesis in the BRCA-deficient context (88).

## Materials and Methods

### Cell culture conditions

An RG37 cell line was derived from SV40-transformed GM639 human fibroblasts and contained the DR-GFP HR reporter (65). RG37, HeLa, U2OS, and U2OS stably expressing RAD52-GFP (kindly provided by Natnael Abate and Michael Hendzel at Laval University (89)) were cultured in DMEM (Gibco) supplemented with 10% FBS. For the CRISPR-LMNA HDR assay, U2OS and HeLa cells were grown in DMEM supplemented with 10% FBS and 1% penicillin/streptomycin.

### DR-GFP HR assay

50,000 cells were seeded 1 d before siRNA transfection, which was carried out using INTERFERin following the manufacturer's instructions (Polyplus-transfection), 40 pmol siRNA: RAD52 (cat #AM16708) was purchased from Ambion, and control and siRAD51 were synthesized by Eurofins (see Table in Supplementary Material). 48 h later, the cells were transfected with the pBASce-HA-I-SceI expression plasmid with Jet-PEI, following the manufacturer's instructions (Polyplus-transfection). Cells were detached 72 h after I-SceI transfection and analyzed by flow cytometry for GFP reporter expression.

### CRISPR-LMNA HDR assay

$1-1.25 \times 10^6$ cells were seeded in 10-cm petri dishes. RAD52 knockdown was performed 4 h later with 50 nM siRNA, using Lipofectamine RNAiMAX (#13778150; Invitrogen), according to the manufacturer's instructions. 24 h after siRNA transfection, the cells were nucleofected with 1 µg of pCR2.1-mRuby2-LMNA (donor), 1 µg of

pX330-LMNA-gRNA2 (guide), and 100 ng of pMAX-GFP (transfection control). Nucleofection was performed using SE Cell Line 4D-Nucleofector X Kit (#V4XC-1024; Lonza) as follows: $1.5 \times 10^6$ cells per condition were pelleted and resuspended in 100 $\mu$l of complete nucleofector solution supplemented with plasmids. Cells were then transferred to a nucleofector cuvette and transfected using the program CM-104 (U2OS) or CN-114 (HeLa) on 4D-Nucleofector X Unit, before reseeding them in 10-cm petri dishes. 48 h after nucleofection, cells were trypsinized and 500,000 cells per condition were seeded in 35-mm petri dishes with a cover glass bottom. The next day, 72 h after nucleofection, cells were incubated with 10 $\mu$g/ml of Hoechst for 30 min at 37°C and then fixed in 4% paraformaldehyde for 10 min at RT. Cells were then visualized using a CellDiscoverer 7 (Zeiss) system, and mRuby2 and GFP expressions were assayed. Image processing and counting were performed using microscope's provided software (Zen). The amount of Ruby2+ cells (HR+ cells) among GFP+ cells (transfected cells) is represented in the graphs as a ratio normalized to the control. Three independent experiments were conducted for each cell line, and at least 500 GFP+ cells were analyzed for each condition and in each replicate. siRNAs are listed in the table (see Supplementary Material).

## PLA

50,000 cells were seeded on coverslips for 24 h, then transfected with 15 or 20 nM siRNA using INTERFERin, following the manufacturer's instructions (Polyplus-transfection). RAD52 siRNA was purchased from Eurogentec, and RAD51 and control siRNAs were synthesized from the Dharmacon ON-TARGET plus SMART pool. Two or three days later, the cells on the coverslip were treated with 200 nM camptothecin (CPT) during 1 h to induce DNA damage, then released during 5 h in the medium. After further washes with PBS, cells were pre-extracted with the CytoSKeleton (CSK) 100 buffer (100 mM NaCl, 300 mM sucrose, 3 mM MgCl$_2$, 10 mM PIPES, pH 6.8, 1 mM ethylene glycol-tetra-acetic acid, 0.2% Triton X-1000, and protease inhibitor cocktail; Roche) during 5 min and fixed with 2% PFA at 37°C for 10 min. Cells were then permeabilized with 0.5% Triton X-100/PBS during 15 min at room temperature. The in situ PLA (Duolink DUO92101) was performed according to the manufacturer's instructions. In brief, after blocking (Duolink blocking solution), coverslips were incubated with the following primary antibodies: anti-RAD51 (Ab-1; Calbiochem, PC130, 1/300) and anti-GFP (Living Color JL-8; Clontech 632381, 1/250). We performed two control conditions, incubation with only one of the primary antibodies and analysis of samples where the expression of the proteins was reduced by silencing (siRAD51 or siRAD52). Coverslips were incubated with secondary antibodies conjugated with PLA probes. All the antibodies were incubated in a humidified chamber at 37°C for 1 h. Ligation of the PLA probes anti-mouse MINUS and anti-rabbit PLUS was performed in a 30-min reaction in a humidified chamber at 37°C, and the signal was amplified with red fluorescence during 1h 40 min in a humidified chamber at 37°C. Samples were mounted in the Duolink mounting medium containing DAPI (blue). Images were acquired randomly using a ZEISS Axio Imager microscope and analyzed with ImageJ. The number of foci per nucleus was counted with CellProfiler software.

## Western blot analysis

Cells were lysed for 30 min at RT in a buffer containing Benzonase (>250 U/ml) in 50 mM Tris–HCl (pH 7.5), 20 mM NaCl, 1 mM MgCl$_2$, and 0.1% SDS, supplemented with a complete mini protease inhibitor (Roche). Proteins (40 $\mu$g) were denatured for 10 min at 55°C, electrophoresed on 9% SDS–PAGE, transferred onto nitrocellulose membranes, and probed with the following specific antibodies: anti-RAD52 (sc-365341; Santa Cruz), anti-RAD51 (PC130; Millipore), and anti-vinculin (Abcam). Immunoreactivity was visualized using an enhanced chemiluminescence detection kit (ECL; Pierce).

## Protein purification

Human RAD51 was purified by CiGEX Platform (CEA, Fontenay-aux-Roses) as follows. His-SUMO-RAD51 was expressed in *E. coli* strain BRL (DE3) pLys. All the protein purification steps were carried out at 4°C. Cells from a 3-liter culture that was induced with 0,5 mM isopropyl-1-thio-ß-D-galactopyranoside overnight at 20°C were resuspended in PBS x1, 350 mM NaCl, 20 mM imidazole, 10% glycerol, 0,5 mg/ml lysozyme, complete protease inhibitor (Roche), and 1 mM 4-(2-aminoethyl)-benzenesulfonyl fluoride (AEBSF). Cells were lysed by sonication, and the insoluble material was removed by centrifugation at 150,000$g$ for 1 h. The supernatant was incubated with 5 ml of Ni-NTA resin (QIAGEN) for 2 h. The mixture was poured into an Econo-Column chromatography column (Bio-Rad), and the beads were washed first with 80 ml W1 buffer (20 mM Tris–HCl, pH 8, 500 mM NaCl, 20 mM imidazole, 10% glycerol, and 0.5% NP-40), followed by 80 ml of W2 buffer (20 mM Tris–HCl, pH 8, 100 mM NaCl, 20 mM imidazole, 10% glycerol, and 1 mM DTT). His-SUMO-RAD51 bound to the beads was resuspended in 8 ml of W2 buffer and incubated with SUMO protease at a ratio 1/80 (W/W) for 16 h. RAD51 without the his-SUMO tag was then recovered into the flow-through and directly loaded onto a HiTrap heparin column (GE Healthcare). The column was washed with W2 buffer, and then, a 0.1–1 M NaCl gradient was applied. Fractions containing purified RAD51 were concentrated and dialyzed against storage buffer (20 mM Tris–HCl, pH 8, 50 mM KCl, 0.5 mM EDTA, 10% glycerol, 1 mM DTT, and 0.5 mM AEBSF) and stored at −80°C.

Human RPA protein was purified on CiGEX Platform (CEA, Fontenay-aux-Roses) as previously described (90). RAD52 and BRCA2 were purified as previously described (91). Protein's purity was verified by SDS–PAGE (see Fig S1). Homogeneity of each purified protein was verified using negative staining TEM.

### Synthesis of the 3'-overhang DNA construction (400 base pairs with a 3' overhang of 1,040 nucleotides)

Two DNA fragments of 1,040 and 400 bp were amplified from the pBR322 plasmid by PCR using *Taq* polymerase and the pairs of primers Cy5-2574$^+$ x biotin-4014$^-$ and biotin-2574$^+$ × 2976$^-$, respectively (see Supplementary Material). Biotinylated PCR products were purified on a MiniQ 4.6/50 ion exchange column (GE Healthcare Life Sciences) and loaded onto a HiTrap Streptavidin HP column (Amersham Biosciences). Purification of the non-biotinylated 400- and 1,040-nucleotide(nt)-long ssDNA was achieved by elution with

80 mM NaOH, neutralized by the addition of HCl 1 M. The ss-dsDNA construction was obtained by annealing of 400- and 1,040-nt ssDNA in equimolar concentrations in molecules in the presence of 1.5 mM $MgCl_2$, then purified on a MiniQ ion exchange column.

### Biochemical assays of DNA–protein complexes for TEM statistical analysis

For RPA-DNA complexes and mediation assay, the ssDNA part of the 3'-overhang DNA substrate was covered with a saturated concentration of RPA as follows. 15 $\mu$M in nucleotides of the DNA substrate was incubated with 0,45 $\mu$M RPA (1 protein per 20 nt of ssDNA) in a buffer containing 10 mM Tris–HCl, pH 7.5, and 50 mM NaCl for 10 min at 37°C. BRCA2 (2–5 nM) and/or RAD52 (0.1–0.5 $\mu$M) were then introduced in the reaction during 15 min at 37°C, and for the mediation assays, 5 $\mu$M RAD51 (1 protein per 3 nt) was added at the same time as the BRCA2/RAD52 partner. In the last experiment, the buffer was supplemented with 2 mM $MgCl_2$, 2 mM $CaCl_2$, 1.5 mM ATP, and 1 mM DTT to allow RAD51 filament formation. For TEM analysis and D-loop assays, RAD51 filaments were formed by incubating 15 $\mu$M in nucleotides of 3'-overhang DNA labeled with Cy5 with 5 $\mu$M RAD51 (1 protein per 3 nt) in a buffer containing 10 mM Tris–HCl, pH 7.5, 50 mM NaCl, 2 mM $MgCl_2$, 2 mM $CaCl_2$, 1.5 mM ATP, and 1 mM DTT for 3 min at 37°C, then adding 0.15 $\mu$M RPA (1 protein per 60 nt) during 15 min. BRCA2 (2–5 nM) or RAD52 (0.1–0.5 $\mu$M) was added to the reaction at the same time as RAD51.

### D-loop in vitro assays and analysis of the DNA–protein and DNA intermediates

In the first step of the reaction, RAD51 filaments were assembled on the 3'-overhang DNA construction as previously described. In the second step, 25 nM in homologous/heterologous dsDNA donor molecules was introduced during 30 min at 37°C. For the homologous donor, pUC19 plasmid was used, whereas PhiX174 RFI was used as heterologous DNA, both purchased from New England Biolabs and purified on a MiniQ ion exchange chromatography column. The D-loop reactions were analyzed by agarose gel electrophoresis. Typically, the D-loop reaction was stopped by adding 0.5 mg/ml Proteinase K, 1% SDS, and 12.5 mM EDTA and incubated overnight at room temperature, and images were obtained by running a 1% TAE agarose gel at 70 V, for 40 min.

### TEM

For DNA–protein complex observation, 0.5 $\mu$l of the different reactions was quickly diluted 120 times in a buffer containing 10 mM Tris–HCl, pH 7.5, 50 mM NaCl, 2 mM $MgCl_2$, and 2 mM $CaCl_2$ and observed by electron microscopy (DNA–protein samples). During one minute, a 5-$\mu$l drop of the dilution was deposited on a 600-mesh copper grid previously covered with a thin carbon film and preactivated by glow discharge in the presence of amylamine (Sigma-Aldrich) (92, 93). Grids were rinsed and positively stained with aqueous 2% (wt/vol) uranyl acetate, dried carefully with a filter paper, and observed in the annular darkfield mode in zero-loss filtered imaging, using a Zeiss 902 transmission electron microscope. Images were captured at a magnification of 85,000× with a

Veleta CCD camera and analyzed with iTEM software (both from Olympus Soft Imaging Solution). For the quantifications, the different populations of molecules were counted on at least two independent experiments with a total of at least 200 molecules counted.

### RAD52 immunolabeling for its TEM detection

To test the presence of RAD52 in the RAD51 filament and in joint-molecule structures, we carried out an immunoaffinity labeling procedure. The DNA–protein complexes were first stabilized with the addition of 0.01% glutaraldehyde and incubated for 10 min at 30°C. Then, the reactions were sequentially incubated for 10 min at 25°C with 3 $\mu$M of a polyclonal RAD52 antibody (GeneTex GTX54722) and 5 $\mu$M of the secondary immunogold antibody (BBI Solutions EM.GFAR5). The reaction was then crosslinked with 0.04% glutaraldehyde (0.05% in final concentration) for its subsequent purification by gel filtration using a superose-6 column (Amersham) to remove the excess of proteins and primary/secondary antibodies. To ensure that the labeling was specific to RAD52 detection, control experiments were performed with pure RAD51 filaments. Samples were visualized by EM in darkfield and brightfield modes.

### Statistical analysis

Statistical analysis was performed with Prism 9 (GraphPad Software). The statistical tests used are indicated in the legends of the Figures. ns = non-significant, * = $P < 0.05$, ** = $P < 0.01$, *** = $P < 0.001$, and **** = $P < 0.0001$.

# Supplementary Information

# Acknowledgements

We greatly thank Natnael Abate and Michael Hendzel, who kindly provided the U2OS RAD52-GFP cell line. We thank the members of our laboratories, especially Valeria Naim, for their helpful comments and suggestions. This work was supported by grants from ANR (FIRE 17-CE12-0015), Paris-Saclay University (MRM and ERM), and CIHR FDN-388879 to J-Y Masson. J-Y Masson is Canada Research Chair in DNA Repair and Cancer Therapeutics. Confocal microscopy observations were supported by the grant CLCB (taxe d'apprentissage Gustave Roussy) to C Basto at the PFIC platform of Gustave Roussy.

## Author Contributions

AA Muhammad: data curation, formal analysis, validation, visualization, and methodology.
C Basto: conceptualization, data curation, formal analysis, validation, investigation, visualization, and methodology.
T Peterlini: resources, data curation, and formal analysis.
J Guirouilh-Barbat: conceptualization, data curation, formal analysis, and investigation.

M Thomas: data curation and formal analysis.

X Veaute: resources.

D Busso: resources.

B Lopez: conceptualization and funding acquisition.

G Mazon: conceptualization, supervision, funding acquisition, project administration, and writing—original draft, review, and editing.

E Le Cam: conceptualization, funding acquisition, methodology, project administration, and writing—original draft, review, and editing.

J-Y Masson: conceptualization, funding acquisition, project administration, and writing—review and editing.

P Dupaigne: conceptualization, data curation, formal analysis, supervision, funding acquisition, validation, investigation, visualization, methodology, and writing—original draft.

## Conflict of Interest Statement

The authors declare that they have no conflict of interest.

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
