## [Reviewer comments · Life Science Alliance]

Life Science Alliance

Human RAD52 stimulates the RAD51-mediated homology search

Ali Muhammad, Clara Basto, Thibaut Peterlini, Josée Guirouilh-Barbat, Melissa Thomas, Xavier Veaute, Didier BUSO, Bernard Lopez, Gérard Mazon, Eric Le Cam, Jean-Yves Masson, and Pauline Dupaigne

DOI: <https://doi.org/10.26508/lsa.202201751>

Corresponding author(s): *Pauline Dupaigne, Inserm*

Review Timeline:

Submission Date:	2022-10-03
Editorial Decision:	2022-11-08
Revision Received:	2023-10-11
Editorial Decision:	2023-11-22
Revision Received:	2023-12-01
Accepted:	2023-12-01

Scientific Editor: *Eric Sawey, PhD*

Transaction Report:

November 8, 2022

Re: Life Science Alliance manuscript #LSA-2022-01751-T

Dr. Pauline Dupaigne
Inserm
IGR
Villejuif 94805
France

Dear Dr. Dupaigne,

Thank you for submitting your manuscript entitled "Human RAD52 stimulates the RAD51-mediated homology search" to Life Science Alliance. The manuscript was assessed by expert reviewers, whose comments are appended to this letter. We invite you to submit a revised manuscript addressing the Reviewer comments.

Thank you for this interesting contribution to Life Science Alliance. We are looking forward to receiving your revised manuscript.

Sincerely,

B. MANUSCRIPT ORGANIZATION AND FORMATTING:

Reviewer #1 (Comments to the Authors (Required)):

Manuscript by Muhammad et al is devoted to characterization of human RAD51 nucleoprotein filament formation and synapsis and the roles RAD52 and BRCA2 proteins are playing in these processes/activities. Authors used electron microscopy and biochemical techniques to answer the long-standing question, simply put, what is the role of RAD52 in HR in humans?

The results represents interesting addition to the field of RAD52 biochemistry by directly looking at the nucleoprotein complexes and, even though conceptually only partially novel, in my opinion, should be published in the journal.

However, Methods requires more elaboration. It is important to always precisely indicate what was done, especially when working with complex reactions containing multiple proteins with various activities. Please see some examples below.

In addition, manuscript requires some "polishing". Again, see some examples below. In particular, References are in very poor shape as of now. The list seems to be alphabetical, but not always. Sometimes Journal name or other info is missing. Refs withing the main text are sometimes misleading.

Certain references are duplicated.
Certain references are duplicated.

Specific comments by page.

Page 3

"DNA gaps"

"it is initiated" - Capital...

"nucleofilament" ???

(Yan et al, 2019)(Hashimoto et al, 2010) - why 4 brackets?

Chen et al 2008a and 2008b are the same references... Same with Jensen et al 2010a,b, same with Tavares...

Page 4

saccharomyces cerevisiae - Capitalize

Page 6

Substrate prep is very poorly described. Is there a ligation step? Or how 400 and 1040 bp/nt DNA fragments been linked?

"BRCA2 (2 to 5 nM) or RAD52 (0,1 to 0,5 µM)"

It is very hard to understand what concntrations have been used in specific experiments. Sometimes this information is lacking. This needs to be corrected throughout the text and/or Figure legends. These are not minor details, but instead are fundamental to the assays used.

Page 7

"controle" -> control

Page 8

(Ms, 1995) - correct this ref (author name, ref details in the referece list)

Page 9

"This specific property of hRAD51 raises the question of whether there is a transient binding of RAD51 to dsDNA" - RAD51

binding to the dsDNA is a well-documented RAD51 protein activity, extensively covered in the literature.

For the order of addition experiments (here and in other experiments), what was the delay, if any, between BRCA2/RAD52 addition?

Page 10

"Greene et al. 2017 (Ma CJ, Kwon Y, Sung P, 2017)." - is this ref to Greene et al, or Ma et al?

Page 12

"Heyer et al. (Piazza et al, 2017)" - again, who's the first author of that paper?

Page 15

"Rad51" -> RAD51; (for the human protein)

"RAD51" -> Rad51 (for the yeast protein)

"desassemble" -> disassemble

Page 16

"wher" -> where

Figs

Fig. 2, "discrete"

What do the cartoons below Fig. 2E represent?

Fig. 5, "independant" -> independent

Reviewer #2 (Comments to the Authors (Required)):

Muhammad et al investigate the role of human RAD52 in homologous recombination (HR). They attempt to elucidate if hRAD52 has a mediator function and whether it can load RAD51 onto RPA coated ssDNA, promoting filament formation and homology search in order to facilitate HR.

Using biochemical assays and transmission electron microscopy (TEM), they show that addition of RAD52 to RPA-ssDNA molecules leads to compaction of the strand. Additionally, as previously elucidated, BRCA2 but not RAD52 can facilitate RAD51 nucleation on RPA coated ssDNA. RAD52 is shown to inhibit RAD51 nucleation by competing with BRCA2 when added in the same mix. The authors suggest these findings support the notion that RAD52 does not have traditional mediator activity. Surprisingly, they also found that if you add RAD52 to pre-formed RAD51-filaments, it leads to the formation of "mixed" filaments with RAD52 interspersed occasionally within the RAD51 filament. They suggest that this phenomenon makes the RAD51 filament more flexible and aids in homology searching. In essence, the authors suggest that BRCA2 is the primary mediator of RAD51 filament formation, while RAD52 acts downstream by the formation of the mixed filaments, aiding in homology search. While the study provides a novel view of RAD52 function, the following points need to be addressed:

- The synaptic complexes assay result in Figure 5F, suggest that complexes can be formed whether the DNA is homologous or not. In this assay, why is the initial contact between the RAD51 coated ssDNA and dsDNA independent of sequence?
- The authors suggest that the RAD52 mixed RAD51 filaments aid in homology search. The observation of mixed filaments of RAD51 interspersed with RAD52 is not proof of "flexible" homology search. The functional conclusion of the TEM observations is not clear.
- The value of the paper would be strengthened by using RAD52 mutants that are deficient in ssDNA binding and RPA or RAD51 interaction to test the underlying mechanism of forming the mixed filaments.
- The authors have previously shown that RAD52 mixed filaments were less susceptible to anti-recombinases like srs2 in yeast and suggest that this may be true in humans as well. For this to be included, it needs to be tested experimentally.
- The authors suggest that RAD52 has a role downstream of BRCA2 in HR. How would you explain the low levels of HR in BRCA2 deficient cells? Are there other proteins that can act as the mediator if not RAD52?
- The authors suggest that RAD52 has no mediator activity based on the TEM imaging results. However, work from other laboratories has suggested that there is a role for RAD52 in RAD51-dependent HR, based primarily on direct repeat HR assays and RAD51 focus formation. The strength of the conclusion should be adjusted accordingly.

Minor issues:

- In Fig 1, the authors have demonstrated the role of RAD52 in HR using the classical DR-GFP assay. They should cite previous papers that have done the same and agree with the data.
- In Fig 2e, the authors should clearly label the X-axis with presence and absence of RAD52.
- In Fig 6c, state the statistical significance (or not) between the heterolog bars.

Reviewer #3 (Comments to the Authors (Required)):

The manuscript by Muhammad et al. utilizes mainly transmission electron microscopy (TEM) to evaluate the impact of RAD52 and BRCA2 on RAD51 filaments assembled on DNA substrates designed to resemble resected DNA (1040 nt's of ssDNA and 400 bp's of dsDNA). The exact role of human RAD52 in homologous recombination or in DNA double-strand break repair in general has remained enigmatic. A prior study (Feng et al. 2011) suggested that RAD52 may play a backup role to BRCA2 in loading RAD51, however, these results have yet to be replicated or followed up by other groups. Most studies to date agree that human RAD52 can anneal complementary ssDNA implying a role in single-strand annealing (SSA) or perhaps second end capture of a broken DNA end. Structural studies have demonstrated that human RAD52 forms oligomeric ring-like structures with DNA wrapped around the inner surface of the ring in a positively charged groove (Stasiak et al. 2000, Parsons et al 2000, Kagawa et al 2002, Lloyd et al., 2005). In this study, the authors propose that human RAD52 is not a mediator of RAD51 loading and nucleation/filament stability, but in fact, inhibits RAD51 loading onto RPA-ssDNA complexes effectively competing with BRCA2 mediator activity.

It seems too preliminary to suggest that RAD52 inhibits RAD51 binding to ssDNA as suggested by this paper without further biochemical investigation. For example, would not any DNA binding protein compete with RAD51 for binding to the DNA substrate? Further, no RAD52 mutants were used to show specificity. Without further evidence, it's difficult to envision how RAD52 could simultaneously block RAD51 assembly on DNA and then somehow promote HDR as suggested by the data in Fig. 1.

Synaptic complexes between invading RAD51-ssDNA filaments and homologous or heterologous DNA substrates are to be expected. Based on its biochemical ability to anneal ssDNA in the presence or absence of RPA, addition of RAD52 would be expected to increase synaptic complexes as shown in the TEM images. It's not clear what the novelty here is.

The authors need to address why addition of BRCA2 does not lead to ssDNA selective binding by RAD51 as has been demonstrated previously for BRC4 and full-length BRCA2 proteins (Carreira et al. 2009, Jensen et al. 2010). These two papers demonstrated that BRCA2 directs RAD51 towards ssDNA and prevents dsDNA binding.

Given the known ability of RAD52 to wrap ssDNA around its ring structure, it's not surprising that the TEM images display reduced DNA lengths in the presence of RAD52. It also doesn't seem surprising that this same activity would reduce the length of RAD51 filaments as less ssDNA would be available for RAD51 to bind. The experiments as shown do not seem to lend any insights as to how RAD52 may or may not stimulate or inhibit HDR. In fact, based on precedence in the literature, it is more likely they act in separate pathways of repair and likely do not act spatially or temporally on the same DNA molecules. Without evidence to the contrary, it's difficult to envision how the authors came up with the model depicted in Fig. 7. Perhaps the authors should also consider a second end capture model for RAD52 function (McIlwraith et al 2008 Mol Cell, Nimonkar et al 2009 PNAS).

Overall, the paper is poorly written with much speculation that seems outside the scope of the question that is being addressed. Interpretations of the TEM images seem subjective and it is difficult to assess, for example, where are the regions of ssDNA and dsDNA located in many of the images, they are not demarcated except where RPA is assumed to bind ssDNA only.

In Fig. 1, the authors show that RAD52 siRNA depletion results in a 40% decrease in HDR activity using the DR-GFP assay. Lok et al. (2013) showed that RAD52 siRNA in U2OS, MCF7, and H1299 cells had no appreciable effect on DR-GFP activity, how do the authors reconcile their results with this previous study? In Fig. 2, RAD52 is shown to decrease the DNA substrate length, which again, is expected based on RAD52's ability to wrap DNA around its ring structure. Fig. 3 suggests that RAD52 interferes with the ability of BRCA2 to support RAD51 loading and filament stability but what is the evidence that BRCA2 and RAD52 compete for the same DNA substrate? In Fig. 4, it is not shown where the proteins are in relation to the single and double-stranded regions of the DNA. As mentioned above, should not BRCA2 direct RAD51 onto ssDNA preventing binding to dsDNA? How is the data in Fig. 4 different from the data in Fig. 3? In Fig. 5, the D-loop gel lanes are not labeled. It is already known that RAD51-ssDNA filaments form synaptic complexes with dsDNA, what is the novelty here? In Fig. 6, how is it known that the images depict multi-invasion species?

The major shortcoming of this study is the lack of in vivo (or cell-based) evidence to support their conclusions, in fact, the prior literature would argue against their conclusions. This would still be OK if the authors devised a rational argument for their specific results but there is no discussion or explanation for their conflicting data. Taken together, the data does not support the

conclusions and further carefully controlled experiments would need to be executed to create a convincing hypothesis that RAD52 indeed plays a role in modulating BRCA2 and RAD51 activity on ds/ssDNA substrates.

Further Specific Issues:

1. More primary articles need to be cited rather than reviews.

Cell culture:

2. Needs more details. In text there is no mention of what cell line is used and why. The cell line is only mentioned in the methods section.

3. The term knockdown is more appropriate in this situation than silencing because there is still a band in the western blot, so the protein level has been reduced but not silenced.

4. My impression is that a cell-based experiment was included to increase the impact and due to criticisms of previous work being solely biochemistry based. However, more cell-based experiments are needed. This one experiment isn't enough. The DR-GFP experiment is highly variable, so it is best used in conjunction with other cell-based experiments to examine HR functionality. To truly show a reduction in RAD52 expression results in decreased HR more experiments are needed. For example, survival in response to DNA damaging agents or γ H2AX levels increased or persist for a greater time with a reduction in RAD52. Figure 1 needs more.

TEM:

5. Some information on the technique of transmission electron microscopy would be useful. It's pros and cons, why it is the best tool for this study.

6. How can you differentiate between proteins bound to the DNA? Example: in figure 3, how do you know the RPA is on the ssDNA and the RAD51 is on the dsDNA? Can you distinguish different proteins using this microscopy?

7. As above, how do you know RPA has been displaced by BRCA2/RAD51?

8. In figure 3D, does the list order represent the order of addition? If so, mention so in text or in figure legend so we know what this image represents.

9. How is it known that RAD52 is in the filament not bound to or on the filament?

Purified proteins:

10. A gel showing all the purified proteins used in this study would be a nice addition.

11. Comment or examination on the activity of the purified proteins is needed. If these proteins are not fully active the results seen in this study could be diminished.

12. BRCA2 concentrations varying (from 1 nM to 2 nM to 5 nM) across experiments. An explanation for why different concentrations were necessary would be informative.

We would like to express our gratitude to the reviewers for their helpful comments and positive feedback that led to experiments of add-value and constructive modifications.

Here enclosed, you can find the responses to each reviewer's comments.

Reviewer #1 (Comments to the Authors (Required)):

Manuscript by Muhammad et al is devoted to characterization of human RAD51 nucleoprotein filament formation and synapsis and the roles RAD52 and BRCA2 proteins are playing in these processes/activities. Authors used electron microscopy and biochemical techniques to answer the long-standing question, simply put, what is the role of RAD52 in HR in humans?

The results represents interesting addition to the field of RAD52 biochemistry by directly looking at the nucleoprotein complexes and, even though conceptually only partially novel, in my opinion, should be published in the journal.

We thank the reviewer for this positive comment.

However, Methods requires more elaboration. It is important to always precisely indicate what was done, especially when working with complex reactions containing multiple proteins with various activities. Please see some examples below.

We agree with the reviewer that some methodological details were missing. We have largely modified this section to add a more precise and detailed description of the methods, and we have also modified figure legends.

In addition, manuscript requires some "polishing". Again, see some examples below. In particular, References are in very poor shape as of now. The list seems to be alphabetical, but not always. Sometimes Journal name or other info is missing. Refs withing the main text are sometimes misleading.

Thanks to the reviewer we have been able to detect errors in the references and we paid attention to improve the wording and overall writing quality.

Certain references are duplicated.
Certain references are duplicated.

Specific comments by page.

Thanks for the detailed comments, we have corrected all the indicated errors and miswording on the manuscript.

For comments below:

Page 6

Substrate prep is very poorly described. Is there a ligation step? Or how 400 and 1040 bp/nt DNA fragments been linked?

The construction of the substrate is achieved in a two steps process: first the purification of 2 ssDNA fragments (400 and 104 nt long), secondly, their annealing in controlled conditions. We paid attention to detail the protocol in the Methods section.

"BRCA2 (2 to 5 nM) or RAD52 (0,1 to 0,5 μM)"

It is very hard to understand what concentrations have been used in specific experiments. Sometimes this information is lacking. This needs to be corrected throughout the text and/or Figure legends. These are not minor details, but instead are fundamental to the assays used.

Again we have modified Methods and figure legends to satisfy this demand.

Page 9

"This specific property of hRAD51 raises the question of whether there is a transient binding of RAD51 to dsDNA" - RAD51 binding to the dsDNA is a well-documented RAD51 protein activity, extensively covered in the literature.

We completely agree with the reviewer, the *in vitro* binding of RAD51 to dsDNA is well documented, but the function of such a property still is intriguing. We have added in the introduction a section describing RAD51 binding to dsDNA and its references.

For the order of addition experiments (here and in other experiments), what was the delay, if any, between BRCA2/RAD52 addition?

When RAD52 and BRCA2 were introduced together into the 'mediation' assay, they were introduced consecutively, one after the other, with an incubation period of 5 minutes between each. We added this precision in the figure legend.

What does the cartoons below Fig. 2E represent?

The cartoon shows how RPA-ssDNA or free ssDNA could be wrapped around RAD52 oligomers and how it could explain the shortening of the complex. We added the explanation in the figure legend.

Reviewer #2 (Comments to the Authors (Required)):

Muhammad et al investigate the role of human RAD52 in homologous recombination (HR). They attempt to elucidate if hRAD52 has a mediator function and whether it can load RAD51 onto RPA coated ssDNA, promoting filament formation and homology search in order to facilitate HR. Using biochemical assays and transmission electron microscopy (TEM), they show that addition of RAD52 to RPA-ssDNA molecules leads to compaction of the strand. Additionally, as previously elucidated, BRCA2 but not RAD52 can facilitate RAD51 nucleation on RPA coated ssDNA. RAD52 is shown to inhibit RAD51 nucleation by competing with BRCA2 when added in the same mix. The authors suggest these findings support the notion that RAD52 does not have traditional mediator activity.

Surprisingly, they also found that if you add RAD52 to pre-formed RAD51-filaments, it leads to the formation of "mixed" filaments with RAD52 interspersed occasionally within the RAD51 filament. They suggest that this phenomenon makes the RAD51 filament more flexible and aids in homology searching. In essence, the authors suggest that BRCA2 is the primary mediator of RAD51 filament formation, while RAD52 acts downstream by the formation of the mixed filaments, aiding in homology search.

We would like to thank Reviewer#2 for his careful and thorough reading of the manuscript.

While the study provides a novel view of RAD52 function, the following points need to be addressed:

- The synaptic complexes assay result in Figure 5F, suggest that complexes can be formed whether the DNA is homologous or not. In this assay, why is the initial contact between the RAD51 coated ssDNA and dsDNA independent of sequence?

It has been shown in different species (bacteria and *Saccharomyces cerevisiae*) that homology search involves a first step of dsDNA probing by the presynaptic nucleofilament, independently of homology, before homology recognition. By using TEM imaging, we have the opportunity to capture transient synaptic complexes formed along the D-loop reaction. In the figure 5, we showed that when we perform this D-loop assay using either a completely homologous or a completely heterologous dsDNA donor (never more than 4 consecutive homologous nucleotides in the sequence), we detect and quantify a large number of synaptic complexes where the nucleofilament is paired with the DNA donor in both cases (homologous - 43 % or heterologous 38 %). The result indicates that the first contacts in the homology search process are independent of homology. Only a fraction of the synaptic complexes will then further form stable D-loop with strand alignment when the homologous sequence is recognized and stably aligned/paired.

- The authors suggest that the RAD52 mixed RAD51 filaments aid in homology search. The observation of mixed filaments of RAD51 interspersed with RAD52 is not proof of "flexible" homology search. The functional conclusion of the TEM observations is not clear.

We appreciate the reviewer's comment; it prompted us to demonstrate quantitatively that 'RAD51-RAD52 filaments are more flexible than pure RAD51 filaments'. Since mixed filaments are not homogeneous, it was not possible to measure and compare their persistence length with pure RAD51 filaments. We decided to measure the angles of curvature, at the break-points of the filaments, where RAD52 accumulates (see the new supplementary figure S3). We observed a significant increase in the number of kinks in the filaments and an increase in the angle from $63,5^\circ \pm 49$ to $90^\circ \pm 69$. The majority of the large angles are associated with the presence of RAD52 (red dots on the graph), showing that RAD52 induced local changes in RAD51 filament flexibility and thus associates with changes in the filament direction. We also show in the manuscript that RAD51-RAD52 filaments are more active in contacting dsDNA. We discuss that in our view the increased flexibility of these mixed filaments is a plausible mechanistic explanation for the increased dsDNA contacts.

- The value of the paper would be strengthened by using RAD52 mutants that are deficient in ssDNA

binding and RPA or RAD51 interaction to test the underlying mechanism of forming the mixed filaments.

We can only agree with the reviewer that the use of mutants would be interesting to further understand the RAD52 molecular mechanisms, and eventually identify separation of function mutants (SSA/HR). We consider that this type of experiments are beyond the scope of the current work, and will be part of future efforts.

- The authors have previously shown that RAD52 mixed filaments were less susceptible to anti-recombinases like srs2 in yeast and suggest that this may be true in humans as well. For this to be included, it needs to be tested experimentally.

We agree with the reviewer that it would be interesting to be able to show this reduction of mixed filaments susceptibility to antirecombinase. Several helicases have been proposed to play the role of yeast's Srs2 in humans. We have tried to study PARI and BLM in the context of our experiments. Nevertheless, and in our hands, we were unable to detect any *in vitro* RAD51 filaments dismantling with these helicases with neither band-shift assays nor TEM experiments. Regardless, we discuss our previous work to contextualize possible functions of the mixed RAD51-RAD52, waiting for confirmatory experiments with helicases yet to identify.

- The authors suggest that RAD52 has a role downstream of BRCA2 in HR. How would you explain the low levels of HR in BRCA2 deficient cells? Are there other proteins that can act as the mediator if not RAD52?

BRCA2 is the main mediator of RAD51 filament formation, and its critical role explains the low level of HR in BRCA2 deficient cells. *In vitro*, if the mediator effect of BRCA2 was clear (figure 3), we nevertheless observed the formation of suboptimal filaments (incomplete) by RAD51 alone, in absence of BRCA2. If these suboptimal filaments exist *in vivo* in BRCA2 deficient cells, they might be not fully efficient in HR, but they may explain low levels of HR, probably more reliant in assisting roles of proteins like RAD52 or RAD59. We cannot also exclude RAD51-independent events mediated by recombinase-like modes of all these accessory factors.

- The authors suggest that RAD52 has no mediator activity based on the TEM imaging results. However, work from other laboratories has suggested that there is a role for RAD52 in RAD51-dependent HR, based primarily on direct repeat HR assays and RAD51 focus formation. The strength of the conclusion should be adjusted accordingly.

In line with the previous comment response, we agree with reviewer 2 that there are some arguments in favor of cooperation of human RAD52 with RAD51 in some contexts. We have rewritten our discussion and added the references accordingly.

Minor issues:

- In Fig 1, the authors have demonstrated the role of RAD52 in HR using the classical DR-GFP assay. They should cite previous papers that have done the same and agree with the data.

yes thanks, it has been corrected

- In Fig 2e, the authors should clearly label the X-axis with presence and absence of RAD52.

done

- In Fig 6c, state the statistical significance (or not) between the heterolog bars.

done

Reviewer #3 (Comments to the Authors (Required)):

The manuscript by Muhammad et al. utilizes mainly transmission electron microscopy (TEM) to evaluate the impact of RAD52 and BRCA2 on RAD51 filaments assembled on DNA substrates designed to resemble resected DNA (1040 nt's of ssDNA and 400 bp's of dsDNA). The exact role of human RAD52 in homologous recombination or in DNA double-strand break repair in general has remained enigmatic. A prior study (Feng et al. 2011) suggested that RAD52 may play a backup role to BRCA2 in loading RAD51, however, these results have yet to be replicated or followed up by other groups. Most studies to date agree that human RAD52 can anneal complementary ssDNA implying a role in single-strand annealing (SSA) or perhaps second end capture of a broken DNA end. Structural studies have demonstrated that human RAD52 forms oligomeric ring-like structures with DNA wrapped around the inner surface of the ring in a positively charged groove (Stasiak et al. 2000, Parsons et al 2000, Kagawa et al 2002, Lloyd et al., 2005). In this study, the authors propose that human RAD52 is not a mediator of RAD51 loading and nucleation/filament stability, but in fact, inhibits RAD51 loading onto RPA-ssDNA complexes effectively competing with BRCA2 mediator activity.

We thank Reviewer #3 for his insightful reading and comments on our manuscript.

It seems too preliminary to suggest that RAD52 inhibits RAD51 binding to ssDNA as suggested by this paper without further biochemical investigation. For example, would not any DNA binding protein compete with RAD51 for binding to the DNA substrate? Further, no RAD52 mutants were used to show specificity. Without further evidence, it's difficult to envision how RAD52 could simultaneously block RAD51 assembly on DNA and then somehow promote HDR as suggested by the data in Fig. 1. In our study, we showed that RAD52 forms discrete complexes associated with RPA-ssDNA, which had not yet been shown and therefore is a novelty. TEM methods historically developed in our lab allow to characterize such ternary complexes in term of assembly and organization.

We provide proof that the presence of a pre-bound RPA but also RAD52-RPA overhangs prevents RAD51 binding and subsequent filament formation. This is a fundamentally different result from what was obtained with the yeast proteins, where Rad52 mediates RPA substitution enabling Rad51 binding and subsequent filament elongation, as we have previously demonstrated using our TEM methods (Esta et al, 2013; Ma et al; 2018). In contrast, when RAD52 was added later in the reaction with RAD51 and RPA, it did not prevent filament assembly, but rather loaded on DNA with RAD51 to form mixed filaments, probably related to the dynamic process of polymerization/depolymerization. We propose that this configuration may be competent to promote HDR.

We absolutely agree with the reviewer that the use of mutants (in particular in the RAD51 interaction domain and also in the N-ter conserved region) would be interesting to go further in the understanding of RAD52 mechanism, with the aim to identify separation of function mutants (SSA/HR), this lies off the scope of this study, and we expect to be able to build in that direction in the near future. We thus limit our conclusions to the available set of data.

Synaptic complexes between invading RAD51-ssDNA filaments and homologous or heterologous DNA substrates are to be expected. Based on its biochemical ability to anneal ssDNA in the presence or absence of RPA, addition of RAD52 would be expected to increase synaptic complexes as shown in the TEM images. It's not clear what the novelty here is.

Given the intrinsic biochemical properties of RAD52, its ability to anneal complementary strands, as the reviewer points out, may suggest a plausible hypothesis of synergistic activity with RAD51 in the homology search process. Yet, we have no clear evidence to proof that hypothesis. An alternative scenario could happen, for instance one can envision that RAD51 by polymerizing on the DNA removes RAD52 or that, DNA wrapping by RAD52 will peel-off any RAD51 filament in close vicinity. Here we show that RAD52 and RAD51 form stable mixed filaments where RAD51 and RAD52 cooperate to contact the dsDNA donor. What we describe limit some of the previous interpretation and thus contributes to advance in the understanding of RAD51 and RAD52 cross-talk.

The authors need to address why addition of BRCA2 does not lead to ssDNA selective binding by RAD51 as has been demonstrated previously for BRC4 and full-length BRCA2 proteins (Carreira et al. 2009, Jensen et al. 2010). These two papers demonstrated that BRCA2 directs RAD51 towards ssDNA and prevents dsDNA binding.

The reviewer highlights an important point here that has resulted in extensive experimental investigations in our side. Pioneering work on BRCA2 by Steve Kowalczykowski's lab showed ssDNA selectivity of the RAD51 filament formation that we did not observe. Experimental conditions may explain these differences. In 2009, Carreira et al. showed that BRC repeats of BRCA2 regulates RAD51 DNA-binding selectivity, so they used truncated forms of BRCA2 in their study. Jensen et al. also showed this selectivity in 2010 by using full length BRCA2 and low RAD51 concentrations (400 nM) but also small DNA substrates. Our experiments have been carried out using longer ss/dsDNA substrates (to mimic the structure and approximate length of a processed DSB *in vivo*), then requiring higher RAD51 concentrations (3 μ M) while only few nM BRCA2 were active in the reactions. It could therefore be possible that this configuration enabled multiple RAD51 nucleation points, thus partially covering the dsDNA. Furthermore, a recent article showed that the BRCA2 C-terminal RAD51-binding segment (TR2) stabilizes RAD51 binding to dsDNA then promoting DNA protection against nuclease activity (Halder et al. Mol Cell 2022). Thus it appears that there is a fine regulation of RAD51 binding to ds/ssDNA by BRCA2 and we think that the preparation of the full length BRCA2 protein but also the reaction conditions could subtly play on the behaviour and interplay between both proteins.

Given the known ability of RAD52 to wrap ssDNA around its ring structure, it's not surprising that the TEM images display reduced DNA lengths in the presence of RAD52. It also doesn't seem surprising that this same activity would reduce the length of RAD51 filaments as less ssDNA would be available for RAD51 to bind. The experiments as shown do not seem to lend any insights as to how RAD52 may or may not stimulate or inhibit HDR. In fact, based on precedence in the literature, it is more likely they act in separate pathways of repair and likely do not act spatially or temporally on the same DNA molecules. Without evidence to the contrary, it's difficult to envision how the authors came up with the model depicted in Fig. 7. Perhaps the authors should also consider a second end capture model for RAD52 function (McIlwraith et al 2008 Mol Cell, Nimmonkar et al 2009 PNAS).

In our study, we show that RAD52 is found linked to RPA-ssDNA complexes or even RAD51-ssDNA in the form of discrete complexes, an observation not made before that we hope the reviewer has been able to appreciate in its novelty. We agree with the reviewer with the lack of "surprise" on the reduced length. The unprecedented property of RAD52 here, in our view, may be its ability to fully coexist in a mixed filament (and not to fulfill a role in helping replace RPA, leaving behind a full and continuous RAD51 filament). The existence of this mixed filament raises new questions on how the properties of such filament are different from a pure RAD51 filament. In the revised manuscript, we have extended our TEM analysis to show that mixed filaments are shorter but also more flexible than pure RAD51 ones (see supplementary figure S3).

The reviewer is right in asking for some evidence of "real" spatial and temporal coexistence of RAD51 and RAD52 (unnecessary for RAD51-independent SSA), this is something we have worked on in this revised version where we show the presence of PLA foci of RAD51 and RAD52 5 hours after DNA damage induction requiring HR repair. The PLA signal requires a proximity of 40 nm between partners, thus strongly suggesting the synchronized co-recruitment of RAD51 and RAD52 and the existence *in vivo* of a situation coherent with the existence of our mixed filaments, even transiently, during HDR. We include however the alternative explanation of second end capture, as the reviewer rightly highlights, an HR step that may also encompass dealing (maybe inhibiting) RAD51 filament formation and invasion from the second end. It will be fascinating to test whether both ends are asymmetrically dealt with during the early steps of HR, something we think we are critically poised to address in our future work.

Overall, the paper is poorly written with much speculation that seems outside the scope of the question that is being addressed.

We are sensitive to this comment, we have made an effort to better communicate in this revised version, and we hope the reviewer would appreciate the attention we made to increase the quality of the writing. We took special care in reducing speculation and stick more to the factual evidences. We can only hope that this revised version can more clearly reach the intended audience, and conveys a clearer message.

Interpretations of the TEM images seem subjective and it is difficult to assess, for example, where are the regions of ssDNA and dsDNA located in many of the images, they are not demarcated except where RPA is assumed to bind ssDNA only.

Our laboratory has a long-standing expertise in TEM imaging recognized worldwide, and has developed its singular approaches dedicated to the observation and characterization of DNA and DNA-protein complexes. More precisely, our TEM methods, dedicated to the visualization of dsDNA as well as ssDNA, combine positive staining and dark field imaging mode then allowing the deployment of molecules on the surface and their imaging with an excellent resolution (around 1 to 2 nm) and contrast. In particular, this approach avoids the use of shadowing techniques, which increase the thickness of ssDNA to around 15-20 nm. We have been applying this method to numerous nucleoprotein complexes for over 30 years. Consequently, interpretations are not subjective, since experiments are systematically observed by more than one scientist and scrupulously quantified. The demarcation between ssDNA and dsDNA is highly visible, since dsDNA is in the form of a worm-like chain, whereas ssDNA is very thin and folded because of secondary structures and its spreading is more random onto the carbon surface of the TEM grid. More recently we have developed a method combining BAC film and positive staining for the characterization of DNA intermediates (ssDNA-dsDNA) but unusual for nucleoprotein complexes (Benureau et al, 2019).

In Fig. 1, the authors show that RAD52 siRNA depletion results in a 40% decrease in HDR activity using the DR-GFP assay. Lok et al. (2013) showed that RAD52 siRNA in U2OS, MCF7, and H1299 cells had no appreciable effect on DR-GFP activity, how do the authors reconcile their results with this previous study?

In this precise type of experiments, we have learnt over time that DR-GFP and similar reporters are extremely sensitive (specially for mild effects) to I-SceI induction conditions. Every single experimental replicate has to be evaluated for its level and synchrony of I-SceI induction. That is the reason why, for each experiment, we have precisely followed the I-SceI expression level to calibrate the timing of our DR-GFP experiment. We know that time-shifted or poorly induced trials, or failure to generate replicates in very reproducible conditions may prevent detection of significant yet “modest” differences on these tests. Just this careful handling could explain the differing results. We were fully aware of this controversial result, which is why we decided to set up a second approach to test with a different reporter and cell line and be completely certain of our results and the conclusion that RAD52 participates in gene conversion events.

In Fig. 2, RAD52 is shown to decrease the DNA substrate length, which again, is expected based on RAD52's ability to wrap DNA around its ring structure. Fig. 3 suggests that RAD52 interferes with the ability of BRCA2 to support RAD51 loading and filament stability but what is the evidence that BRCA2 and RAD52 compete for the same DNA substrate?

TEM allows us to follow reactions over time by freezing equilibrium states on TEM grids, and to perform statistical analyzes of the populations of molecules/complexes present in the reaction. In this experiment (figure 3), we quantified the number of complete filaments formed. We showed that in the conditions where RPA was first bound to the ssDNA, BRCA2 was necessary to allow RAD51 nucleation and assembly on DNA, to form complete filaments. When RAD52 was added in the reaction, filaments were not properly/completely formed. This showed that RAD52 inhibits the BRCA2 mediator role for RAD51 filament assembly. We then suggested (and not asserted) that RAD52 by tightly binding and compacting RPA-ssDNA may compete with RAD51 then avoiding RAD51 nucleation on ssDNA.

In Fig. 4, it is not shown where the proteins are in relation to the single and double-stranded regions of the DNA. As mentioned above, should not BRCA2 direct RAD51 onto ssDNA preventing binding to dsDNA? How is the data in Fig. 4 different from the data in Fig. 3?

We cannot clearly distinguish ssDNA from dsDNA as they are covered by RAD51.

The data presented in figure 4 are totally different from those in figure 3 as RPA was not first incubated with the substrate in saturating concentrations, then avoiding RAD51 nucleation on ssDNA. The objective of this experiment was to see the effect of RAD52 and BRCA2 upon RAD51 filament elongation. As mentioned above, in our conditions no selectivity for RAD51 binding to ssDNA was observed.

In Fig. 5, the D-loop gel lanes are not labeled. It is already known that RAD51-ssDNA filaments form synaptic complexes with dsDNA, what is the novelty here?

These experiment and data are new because we were able for the first time to directly observe and characterize the synaptic complexes formed along the D-loop reaction. We revealed the formation of a high rate of RAD51-mediated synaptic complexes.

In Fig. 6, how is it known that the images depict multi-invasion species?

One nucleofilament is in contact with more than one dsDNA donor (2 or 3 supercoiled plasmids)

The major shortcoming of this study is the lack of *in vivo* (or cell-based) evidence to support their conclusions, in fact, the prior literature would argue against their conclusions. This would still be OK if the authors devised a rational argument for their specific results but there is no discussion or explanation for their conflicting data. Taken together, the data does not support the conclusions and further carefully controlled experiments would need to be executed to create a convincing hypothesis that RAD52 indeed plays a role in modulating BRCA2 and RAD51 activity on ds/ssDNA substrates.

We have taken the final comment of reviewer 3 as motivation for an additional set of *in vivo* experiments to back our mostly biochemical and *in vitro* approached conclusions. As for the prior literature precedence, we have strengthened our rationale for our conclusions.

Further Specific Issues:

1. More primary articles need to be cited rather than reviews.

We did an effort to cite the primary articles

Cell culture:

2. Needs more details. In text there is no mention of what cell line is used and why. The cell line is only mentioned in the methods section.

We have added cell lines in the results section.

3. The term knockdown is more appropriate in this situation than silencing because there is still a band in the western blot, so the protein level has been reduced but not silenced.

ok we corrected.

4. My impression is that a cell-based experiment was included to increase the impact and due to criticisms of previous work being solely biochemistry based. However, more cell-based experiments are needed. This one experiment isn't enough. The DR-GFP experiment is highly variable, so it is best used in conjunction with other cell-based experiments to examine HR functionality. To truly show a reduction in RAD52 expression results in decreased HR more experiments are needed. For example, survival in response to DNA damaging agents or γ H2AX levels increased or persist for a greater time with a reduction in RAD52. Figure 1 needs more.

We have done a real effort to address this comment. In that way we have used a second construction to test HR in cells: the CRISPR-LMNA assay, then we extended the experiments to other cell lines. In

parallel we have carried out the PLA assay in cells treated with Camptothecin and we were able to show the co-recruitment of RAD51 and RAD52, 5 hours after treatment.

TEM:

5. Some information on the technique of transmission electron microscopy would be useful. It's pros and cons, why it is the best tool for this study.

We have added some arguments for the use of TEM approach in this study: 'We have observed by using positive staining EM method in dark field imaging mode the sequential recruitment to the DNA substrate of RPA, RAD52 and BRCA2 as well as RAD51 to the DNA substrate in order to better understand the interplay between these different actors in the assembly of the RAD51 filament, its architecture and its activity. TEM allowed us to directly observe and characterize the molecular features of the transient DNA-protein intermediates generated at different time points during the nucleofilament formation and homologous pairing reactions.'

6. How can you differentiate between proteins bound to the DNA? Example: in figure 3, how do you know the RPA is on the ssDNA and the RAD51 is on the dsDNA? Can you distinguish different proteins using this microscopy?

We've been working with recombinases for a long time and have learned to localize them very precisely in a complex with DNA: recombinases polymerize and form regular helical structures, which we identify very clearly with our approaches by zooming in on the molecules. On the other hand, RPA-like SSB proteins only have affinity for ssDNA (not for dsDNA) and also form recognizable complexes, very different from RAD51 filaments in that they are much less thicker and shiny (Dupaigne et al. 2018 *Methods Mol Biol.*).

7. As above, how do you know RPA has been displaced by BRCA2/RAD51?

Same explanation. RAD51 filaments assembled in presence of BRCA2 are long and continue (as seen on negative staining TEM images) but we agree that we cannot exclude the possibility of some punctual and very local binding of RPA on ssDNA.

8. In figure 3D, does the list order represent the order of addition? If so, mention so in text or in figure legend so we know what this image represents.

Yes thank you, we have done this.

9. How is it known that RAD52 is in the filament not bound to or on the filament?

We cannot completely exclude that RAD52 is not bound onto the filament. But the RAD51-RAD52 mixed filament were only observed when RAD52 was added in the reaction at the same time as RAD51. When RAD52 was introduced once the filament had been preassembled, we did not observe these discrete complexes and no RAD52 was immunolocalized on/in the filament.

Purified proteins:

10. A gel showing all the purified proteins used in this study would be a nice addition.

We have added a supplementary figure 1 showing gels.

11. Comment or examination on the activity of the purified proteins is needed. If these proteins are not fully active the results seen in this study could be diminished.

Homogeneity of each purified protein is verified using TEM negative staining. We also checked that they were not contaminated with nucleases. Then for RAD51, its activity was tested in the strand exchange and D-loop assays, as it is known that the optimal activity is obtained for a stoichiometry of 1 protein per 3 nucleotides. RPA activity was tested for its affinity for ssDNA. The activity test for BRCA2 has been the mediation assay. Finally, concerning RAD52 we verified it well formed ring structures in solution (using TEM negative staining) and we also tested its activity in SSA (Single strand annealing) assays (see So A, Dardillac E, Muhammad A, Chailleux C, Sesma-Sanz L, Ragu S, Le Cam E, Canitrot Y, Masson JY, Dupaigne P, Lopez BS, Guirouilh-Barbat J. *NAR* 2022).

12. BRCA2 concentrations varying (from 1 nM to 2 nM to 5 nM) across experiments. An explanation for why different concentrations were necessary would be informative.

By testing different concentrations, we wanted to try and reveal a BRCA2 concentration-dependent effect.

November 22, 2023

RE: Life Science Alliance Manuscript #LSA-2022-01751-TR

Dr. Pauline Dupaigne
Inserm
Gustave Roussy Cancer Research Campus
39 rue Camille Desmoulins
Villejuif 94805
France

Dear Dr. Dupaigne,

Thank you for submitting your revised manuscript entitled "Human RAD52 stimulates the RAD51-mediated homology search". We would be happy to publish your paper in Life Science Alliance pending final revisions necessary to meet our formatting guidelines.

- please upload your main and supplementary figures as single files
- please add a Running Title to our system
- please add the Twitter handle of your host institute/organization as well as your own or/and one of the authors in our system
- please make sure the author order in your manuscript and our system match; -the full name (middle names as initials) of each author should be given on the title page
- please consult our manuscript preparation guidelines <https://www.life-science-alliance.org/manuscript-prep> and make sure your manuscript sections are in the correct order
- please be sure to add all Authors in the Author Contribution section in the manuscript file
- please add your main, supplementary figure, and table legends to the main manuscript text after the references section
- we encourage you to revise the figure legend for Figure 5 such that the figure panels are introduced in an alphabetical order
- please upload your Table in editable .doc or excel format;
- please add callouts for Figures 1C, F; 4I,J; S1A-D; S3C-F; S4A-C and table S1 to your main manuscript text
- the activities that were selected in our system for Gérard Mazon do not qualify for authorship. Please either update the contributions, or let us know if this author should be removed from the author list. This author also seems to be referred to as GMB in the Author Contributions section, which does not match their initials based upon the displayed name.

FIGURE CHECKS:

- please add sizes next to all blots
- please provide the original blot used in Figure 5B as Source Data

A. FINAL FILES:

B. MANUSCRIPT ORGANIZATION AND FORMATTING:

Sincerely,

Reviewer #2 (Comments to the Authors (Required)):

The rebuttal to the previous review has addressed many of the issues raised. The model is now that RAD52 provides breaks in the RAD51 filament, introducing some flexibility in the filament, perhaps making homology search easier to perform. RAD52 does "interact" with RAD51, based on new PLA data, which is consistent with other observations. RAD52 does not stimulate D-loop formation but does stimulate the number of synaptic complexes observed by TEM, including junctions without sequence homology.

The data support the model that RAD52 can create "nodes" along a RAD51 filament, which facilitates synaptic complexes, but not long homology alignment, which clearly needs BRCA2. Although, some of the published data support an independence of RAD52 function from BRCA2, the current model suggests an interdependence for optimal homologous recombination. The assay may determine how these genetic manipulations are observed.

Overall, there are enough interesting observations to support publication.

December 1, 2023

RE: Life Science Alliance Manuscript #LSA-2022-01751-TRR

Dr. Pauline Dupaigne
Inserm
Gustave Roussy Cancer Research Campus
114 rue Edouard Vaillant
Villejuif 94805
France

Dear Dr. Dupaigne,

Thank you for submitting your Research Article entitled "Human RAD52 stimulates the RAD51-mediated homology search". It is a pleasure to let you know that your manuscript is now accepted for publication in Life Science Alliance. Congratulations on this interesting work.

DISTRIBUTION OF MATERIALS:

Again, congratulations on a very nice paper. I hope you found the review process to be constructive and are pleased with how the manuscript was handled editorially. We look forward to future exciting submissions from your lab.

Sincerely,
